# Optimizing metaproteomics database construction: lessons from a study of the vaginal microbiome

Elliot M. Lee,[1,2] Sujatha Srinivasan,[1] Samuel O. Purvine,[3] Tina L. Fiedler,[1] Owen P. Leiser,[3] Sean C. Proll,[1] Samuel S. Minot,[1] Brooke L. Deatherage Kaiser,[3] David N. Fredricks[1,2]

**ABSTRACT**  Metaproteomics, a method for untargeted, high-throughput identification of proteins in complex samples, provides functional information about microbial communities and can tie functions to specific taxa. Metaproteomics often generates less data than other omics techniques, but analytical workflows can be improved to increase usable data in metaproteomic outputs. Identification of peptides in the metaproteomic analysis is performed by comparing mass spectra of sample peptides to a reference database of protein sequences. Although these protein databases are an integral part of the metaproteomic analysis, few studies have explored how database composition impacts peptide identification. Here, we used cervicovaginal lavage (CVL) samples from a study of bacterial vaginosis (BV) to compare the performance of databases built using six different strategies. We evaluated broad versus sample-matched databases, as well as databases populated with proteins translated from metagenomic sequencing of the same samples versus sequences from public repositories. Smaller sample-matched databases performed significantly better, driven by the statistical constraints on large databases. Additionally, large databases attributed up to 34% of significant bacterial hits to taxa absent from the sample, as determined orthogonally by 16S rRNA gene sequencing. We also tested a set of hybrid databases which included bacterial proteins from NCBI RefSeq and translated bacterial genes from the samples. These hybrid databases had the best overall performance, identifying 1,068 unique human and 1,418 unique bacterial proteins, ~30% more than a database populated with proteins from typical vaginal bacteria and fungi. Our findings can help guide the optimal identification of proteins while maintaining statistical power for reaching biological conclusions.

**IMPORTANCE**  Metaproteomic analysis can provide valuable insights into the functions of microbial and cellular communities by identifying a broad, untargeted set of proteins. The databases used in the analysis of metaproteomic data influence results by defining what proteins can be identified. Moreover, the size of the database impacts the number of identifications after accounting for false discovery rates (FDRs). Few studies have tested the performance of different strategies for building a protein database to identify proteins from metaproteomic data and those that have largely focused on highly diverse microbial communities. We tested a range of databases on CVL samples and found that a hybrid sample-matched approach, using publicly available proteins from organisms present in the samples, as well as proteins translated from metagenomic sequencing of the samples, had the best performance. However, our results also suggest that public sequence databases will continue to improve as more bacterial genomes are published.

**KEYWORDS**  metaproteomics, proteomics, metagenomics, protein databases, database optimization, microbiome

Address correspondence to David N. Fredricks, dfredric@fredhutch.org.

Brooke L. Deatherage Kaiser and David N. Fredricks contributed equally to this article. More than one author was listed as senior author. Authorship order was selected based on increasing seniority with equal contributions.

David Fredricks and Tina Fiedler jointly hold a patent for diagnosing bacterial vaginosis.

See the funding table on p. 24.

Interactions between the microbiota and their host, and among different microbes within that community, can have a large impact on host health (1, 2). Omics techniques have enabled the elucidation of composition and functional characteristics of these communities (3–7). Metaproteomics is an especially useful tool as it provides functional information from a complex sample potentially containing many different genera and species (8). Most often, metaproteomic analysis is performed by digesting proteins from a sample with trypsin and analyzing the resultant peptides by high resolution tandem mass spectrometry (9). By identifying the proteins which yielded these peptides, it is possible to make functional inferences about the community at the time of sampling. Further, semiquantitative data from spectral counts associated with peptides allow researchers to draw conclusions about functional differences under varying conditions (10). Peptides can often be associated with specific taxa, providing a method to determine what organisms may be responsible for specific functions (11). While metaproteomics is a powerful technique, it generates a much smaller volume and depth of data compared to other omics approaches such as metagenomics and metatranscriptomics, and additional research is needed to improve the efficiency of metaproteomic analysis (12–15).

Peptides obtained from tryptic digestion of proteins in a sample are identified by comparing them to protein sequences in a reference database (16–18). A search algorithm digests the user-supplied sequences *in silico* and generates theoretical mass spectra of database peptides to compare against experimental spectra. The algorithm determines the best match for each experimental spectrum and assigns each resultant peptide-to-spectrum match (PSM) a significance value. At the same time, the algorithm compares the sample spectra to a decoy database filled with random protein sequences, and significant identifications against this database are used to calculate the FDR of the search. Both significance of an individual PSM as well as FDR of the search are used to calculate a $q$ value, the metric used to determine whether a PSM is of sufficiently high-quality to include in further analysis. Because including more proteins in a database increases the chances of finding a match for a given spectrum, but increasing database size also increases the likelihood of a false match; database construction is a critical and complex step in any proteomic analysis (19). Few studies have investigated how the composition of a protein database influences the amount of usable data generated by a search, and most studies conducted thus far focus on the gut microbiome or simple artificial communities (20–23). Here, we analyzed CVL samples from 29 women to characterize proteins from the vaginal microbiome. Although the vaginal microbiome is complex, its communities tend to be much less diverse than the gut microbiome (24–26). We compared the performance of protein databases that varied in microbial diversity as well as the source of sequence data, whether from public repositories or from shotgun metagenomic sequencing of the samples. Focused, sample-matched databases generated the highest number of statistically significant identifications of sample spectra from the dataset. Combining databases that contained public sequence data and sample-matched shotgun metagenomic sequencing increased performance over each individual strategy. Finally, we provide a systematic approach to guide building custom protein databases for future metaproteomic studies to optimize results.

## MATERIALS AND METHODS

### Study population and sample collection

The parent study enrolled 242 women from the Public Health, Seattle and King County Sexually Transmitted Diseases Clinic (STD clinic) between September 2006 and June 2010 (26). The study was approved by the Institutional Review Board at the Fred Hutchinson Cancer Research Center and all study participants provided written informed consent. Vaginal fluid samples were collected for molecular characterization, Gram-staining, pH, saline microscopy, and potassium hydroxide preparation. BV was diagnosed using Amsel clinical criteria (27) and confirmed by Gram-stain using the Nugent and Hillier method

(28). BV diagnosis was the same by both Nugent score and Amsel criteria for all 29 participants whose samples were analyzed by mass spectrometry. CVL was collected by instilling 10 mL sterile saline into the vagina using a needleless syringe, and walls of the vagina were washed to remove any adherent cells. After ~1 min, the lavage fluid was aspirated and stored at −80°C. CVL was used for proteomic analyses. A random cross-sectional case-control set of 20 samples from women with BV and 10 samples from women without BV were selected for proteomics. As there is greater bacterial heterogeneity in BV, we used a 2:1 case-to-control ratio for sample selection to have a better representation of women with BV. One sample could not be analyzed leaving 29.

## Swab preparation and DNA extraction from vaginal swab samples

Vaginal swabs collected from study participants were frozen at −80°C until processing. Swabs were prepared for extraction by placing the swab tip in a tube with 500 μL filtered 0.9% saline (100K MWCO). Swabs were vortexed for 1–2 min and then removed. Vaginal fluid from the swab tip was centrifuged at 14,000 rpm for 10 min at 4°C to pellet cells. Genomic DNA (gDNA) was extracted from pellets using the BiOstic Bacteremia DNA Isolation Kit (Qiagen, Germantown, MD, USA). DNA was eluted in 150 μL buffer. DNA extraction controls (blank swabs without contact with human mucosa) were included for every 15 samples to assess contamination from extraction reagents or collection swabs. DNA was stored at −80°C until sequencing.

## Broad-range PCR and sequencing for microbiota characterization

Total bacterial DNA concentrations (16S rRNA gene copies) were measured using a qPCR assay targeting the V3-V4 region of the 16S rRNA gene (26). Samples were evaluated for the presence of PCR inhibitors using a qPCR assay targeting a segment of exogenously added jellyfish DNA, and inhibition was defined as a delay in the threshold of >2 cycles compared to no-template controls (29). Relative abundances of bacterial taxa sequence reads were measured using broad-range PCR targeting the V3-V4 region of the 16S rRNA gene and sequencing on the Illumina MiSeq instrument (Illumina, San Diego, CA, USA) (30). Raw sequence reads were demultiplexed using the Illumina MiSeq's onboard software. Demultiplexed reads were processed using *barcodecop* v0.4.1 (Hoffman NG. barcodecop. 2019. https://github.com/nhoffman/barcodecop) to enforce barcode quality using the default setting and to ensure exact barcode matches to the forward and reverse reads. The *DADA2* package version 1.6.0 was used for quality filtering, read trimming, error correction and dereplication, paired-end assembly, and chimera removal resulting in a list of unique sequence variants (31). Sequence variants were classified using the phylogenetic placement tool *pplacer* (32) and a curated set of vaginal bacteria (25). Sequence reads are available from the NCBI Short Read Archive (Accession number PRJNA881379).

## Metagenomics library preparation and sequencing

gDNA from vaginal samples was quantified via Qubit fluorometer and Quant-iT dsDNA Assay Kit of high sensitivity (Life Technologies-Invitrogen, Carlsbad, CA, USA). Sequencing controls included a bacterial mock community (ATCC MSA-1003, Manassas, VA, USA) and a bacterial isolate DNF00720 *Fannyhessea vaginae*. Sequencing libraries were prepared from 250 pg gDNA with a quarter reaction workflow using the Nextera XT Library Prep Kit (Illumina, San Diego, CA, USA). Libraries were pooled by volume and post-amplification cleanup was performed with 0.8X Agencourt AMPure XP beads (Beckman Coulter, Indianapolis, IN, USA). The library pool size distribution was validated using the Agilent High Sensitivity D5000 ScreenTape run on an Agilent 4200 TapeStation (Agilent Technologies, Inc., Santa Clara, CA, USA). Additional library QC and cluster optimization was performed using Life Technologies-Invitrogen Qubit 2.0 Fluorometer (Life Technologies-Invitrogen, Carlsbad, CA, USA). Sequencing was performed on a NovaSeq 6000 S1-300 flowcell (Illumina, San Diego, CA, USA). Image analysis and

base calling were performed on board the NovaSeq 6000 with Real Time Analysis v3.4.4 software. Generation of Fastq files was performed with Illumina's bcl2fastq 2.20 Conversion software. The *geneshot* workflow was used to process Fastq files (33). This workflow fed data into metaSPAdes for assembly and Prokka for annotation of bacterial genes (34, 35).

## Proteomic sample preparation

Cervical lavage (CVL) samples were reduced, denatured, and digested to peptides for mass spectrometry analysis. First, 105 mg of solid urea was added to 250 µL of CVL to yield a final concentration of 7 M. Dithiothreitol was then added to a final concentration of 5 mM, and the sample was incubated for 30 min at 60°C with gentle shaking (300 rpm) in a thermomixer. Following incubation, 2.25 mL of 50 mM ammonium bicarbonate was added to each tube. USB brand trypsin was resuspended to 1 µg/µL in acetic acid, and 5 µL was added to each sample. Samples were incubated at 37°C with gentle shaking (300 rpm) overnight. Digested peptide samples were centrifuged for 5 min at 12,000 × *g* to pellet any remaining solid debris and the supernatant was subjected to solid phase extraction (SPE). SPE cartridges (Phenomenex; Strata C18-T (55 µm, 140 Å), 100 mg/1 mL, cat # 8B-S004-EAK) were loaded into the vacuum manifold, and samples were processed according to the manufacturer's recommendation. Briefly, cartridges were conditioned with 1 mL methanol and washed with 1 mL 0.1% trifluoroacetic acid (TFA) in water. The sample was added to the cartridge, followed by a wash with 1 mL 5% acetonitrile/95% 0.1% TFA in water. Finally, the sample was eluted with 1 mL 80% acetonitrile and 20% 0.1% TFA in water. Samples were concentrated to near dryness using a SpeedVac and resuspended in 30 µL 0.1% TFA water. A BCA assay (Pierce) was performed to determine peptide concentration. Samples were diluted to 0.1 µg/µL and stored at −80°C until MS analysis.

## LC-MS/MS analysis

A Waters nano-Acquity dual pumping UPLC system (Waters, Milford, MA, USA) was configured for online trapping of a 5 µL injection at 5 µL/min with reverse-flow elution onto the analytical column at 300 nL/min. Columns were packed in-house using 360 µm o.d. fused silica (Polymicro Technologies Inc., Phoenix, AZ, USA) with 2 mm sol-gel frits for media retention and contained Jupiter C18 media (Phenomenex, Torrence, CA, USA) in 5 µm particle size for the trapping column (150 µm i.d. × 4 cm long), with 3 µm particle size for the analytical column (75 µm i.d. × 70 cm long). Mobile phases consisted of (A) 0.1% formic acid in water and (B) 0.1% formic acid in acetonitrile with the following gradient profile (min, %B): 0, 1; 2, 8; 20, 12; 75, 30; 97, 45; 100, 95; 110, 95; 115, 1; 150, 1. MS analysis was performed using a Velos Orbitrap mass spectrometer (Thermo Scientific, San Jose, CA, USA) outfitted with a custom electrospray ionization (ESI) interface. Electrospray emitters were custom made by chemically etching 150 µm o.d. × 20 µm i.d. fused silica (36). The heated capillary temperature and spray voltage were 350°C and 2.3 kV, respectively. Data was acquired for 100 min after a 15 min delay from when the gradient started. Orbitrap spectra (AGC 1 × 106) were collected from 400 to 2,000 m/z at a resolution of 60 k followed by data-dependent ion trap MS/MS (collision energy 35%, AGC $1 \times 10^4$) of the 10 most abundant ions. A dynamic exclusion time of 45 s was used to discriminate against previously analyzed ions using a 0.55 to 1.55 Da mass window. Each sample was analyzed in two separate replicates.

## Database construction for identification of peptide sequences

Seven different protein sequence databases were built for the identification of peptides. All databases included human protein sequences from Swiss-Prot (release 2019_02) and 16 common contaminants including human keratins that could be introduced during sample processing and trypsins that could be left over from sample preparation. The Global database consisted of all bacterial, fungal, and *Trichomonas vaginalis*

sequences available on NCBI RefSeq (RefSeq release 97), downloaded on 19 June 2020. A separate 16S_Sample-Matched database was built for each sample and included all RefSeq sequences for bacterial taxa present in the sample at >0.1% abundance according to 16S rRNA gene sequencing, except BVAB2 for which there were no available genomes at the time of publication. The 16S_Pooled database was constructed using all bacterial protein sequences used to build the 16S_Sample-Matched databases, as well as all RefSeq protein sequences for *T. vaginalis, Chlamydia trachomatis, Neisseria gonorrhoeae*, and the common vaginal fungi *Alterna alternata, Candida albicans, Candida glabrata, Candida tropicalis, Pichia kudravzevii,* and *Saccharomyces cerevisiae* (RefSeq release 99) downloaded on 19 June 2020. For the 16S_Pooled and 16S_Sample-Matched databases, sequences for individual microbes were downloaded from RefSeq. The 16S_Reference databases also only included RefSeq proteins for bacterial taxa present in the sample at >0.1% abundance, but only included proteins translated from the genome of the reference strain for that species, as listed on the NCBI website (37). After shotgun metagenomic sequencing was performed on each sample, a sample-matched Shotgun_Sample-Matched database was built using only the translated bacterial proteins identified in that sample. A single Shotgun_Pooled database was also built by pooling together the translated bacterial proteins from all 29 samples. Sample-matched Hybrid_Sample-Matched databases were built using all the proteins from a sample's 16S_Sample-Matched and Shotgun_Sample-Matched databases. All nonhuman FASTA sequences were normalized before being incorporated into databases with the SpeciesSeqPrepper.py program. *Gardnerella* species were delineated using the same groups described in Vaneechoutte et al. (38). Recently published *Gardnerella* genomes were submitted to the DSMZ genome-to-genome distance calculator (39), and a cutoff of 70% similarity was used to determine new species.

## Database construction for comparison of databases with additional strains versus additional species

A baseline database was built using human Swiss-Prot sequences (release 2019_02) and 16 common contaminants, as well as protein sequences from the strains of 10 common vaginal bacteria that provided the most PSMs in a search of the 16S_Sample-Matched databases. These strains were *Gardnerella swidsinskii* GS10234, *Gardnerella leopoldii* UMB0912, *Gardnerella vaginalis* UMB0411, *Gardnerella vaginalis* DNF01149, *Gardnerella piotii* UGhent 18.01, *Lactobacillus iners* SPIN 1401G, *Megasphaera lornae, Prevotella timonensis* DSM 22865, Candidatus *Lachnocurva*, and *L. crispatus* JV-V01. Databases were built by adding protein sequences from 20, 40, 60, 80, or 100 randomly chosen genomes. For the Additional Strains databases, these genomes were chosen from 103 *L. crispatus* genomes available in RefSeq release 97. For the Additional Species databases, these genomes were chosen from 103 random bacterial species of different genera assembled to the scaffold level as part of the Human Microbiome Project (RefSeq release 99). Three separate databases were built and tested for each size.

## Database searching

Approximately half of the bacterial peptides identified in searches of the first replicates were not identified in the second replicates and vice-versa, so both replicates for each sample were combined for further proteomic analysis. Peptide identification was performed with MS-GF+ (v2019.01.22). Isotope error range was −1 to 2, maximum modifications per peptide were set to 3, and peptides were only considered if they were at least partially tryptic. The Nextflow workflow manager (v19.07.0) was used to parallelize and automate the data analysis pipeline.

A two-step database search method was used to maximize data from each sample (40). Briefly, after an initial database search was completed, every protein matched as part of that peptide search was recorded, regardless of statistical significance. A subset protein database was then constructed using only the sequences of the proteins identified in the initial search. A second search was performed using these subset

databases, and the results of this search were used for downstream analysis. A decoy database was created and searched concurrently to calculate FDR and $q$ values.

## Sample selection for comparison of Global against other database types

Because the Global database was 100 x larger and more expensive to search against compared to the next largest database (16S_Pooled), a subset of samples was selected for comparison of the Global, 16S_Pooled, 16S_Sample-Matched, Shotgun_Pooled, Shotgun_Sample-Matched, and Hybrid_Sample-Matched databases. In our dataset, 31% of samples were BV−, so two BV− samples and four BV+ samples were chosen at random from the groups. Spectra in these samples were then searched against all seven database types as described above and the results were used to compare the database types.

## Determination of cost and computing requirements for searching databases

The Nextflow computational pipeline was run with the "-with-report" flag so an .html report would be generated for each execution. The number of CPU hours required to run each search was then totaled from this document, and cost to run each search was calculated based on Amazon Web Services (AWS) published cost per hour for a c6g.4xlarge spot instance (as of March 2022).

## Proteomic data analysis

Search results were analyzed in JupyterLab v1.1.4 (v1.1.4) using Python (v3.6.4). FDR was limited by only including peptide-spectrum matches (PSMs) with a Q-value <0.01. Spectra were classified in a hierarchical manner. If the spectra matched any decoy proteins, it was flagged as a decoy identification and discarded. If the spectra matched any contaminant proteins, it was flagged as a contaminant identification and similarly discarded. Otherwise, if the spectrum matched any eukaryotic proteins, it was categorized preferentially as human, then fungal, then *Trichomonas* if it only matched *Trichomonas* proteins. Finally, the spectrum was characterized as bacterial if it only matched bacterial proteins. Peptides were included in further analysis if they met one of the following conditions: (i) MS-GF+ assigned the PSM a spectral probability value <1E-15. (ii) The peptide was one of two unique peptides that matched the same protein. To calculate the relative abundance of certain human proteins, the total number of PSMs attributed to a protein was divided by the total number of human PSMs. These data were then log transformed (base 2) for statistical analysis. Statistical tests were performed using the scipy (v1.3.1) "stats" package. Mann-Whitney U tests were used to assess differences in relative abundance of specific proteins across samples.

## Comparison of database performance—PSMs generated

For each sample, the number of significant PSMs was determined for both human and bacterial spectra as described above. Comparisons were made between databases using the scipy (v1.3.1) "stats" package to perform Wilcoxon Signed-Rank tests on each combination of databases. A database was considered to generate significantly more human or bacterial PSMs than the other if the test reported $P < 0.01$. Relative identification rates for human or bacterial PSMs were determined by first calculating the average number of significant PSMs of the given type identified in the sample when it was searched against the 16S_Pooled, 16S_Sample-Matched, Shotgun_Pooled, Shotgun_Sample-Matched, and Hybrid_Sample-Matched databases. The number of significant PSMs found in that sample by a single database search was then divided by the average significant PSMs for the sample across all databases. This proportion was the relative identification rate.

## Functional analysis

Functional annotations for human and bacterial proteins were separately gathered by querying identified protein sequences against the eggNOG-Mapper web server (v1.0.3)

(41, 42). For PSMs that matched multiple proteins, only the first protein match was queried. Each GO number as assigned by eggNOG-Mapper was then given a spectral count in each sample by totaling the number of PSMs associated with its protein(s). The spectral count of each functional annotation was then divided by the total number of human or bacterial PSMs in the sample, then log transformed (base 2) and tested for statistical significance as described above.

## Taxonomic analysis of proteomic data

Taxonomic assignment of peptides was performed using the taxonomic information attached to protein sequences in each database. All potential protein hits in the database were identified, and all species encoding those proteins were noted.

## Quantification of unique proteins

The number of unique human and bacterial proteins identified by each database search was determined by first applying the filtering criteria described above to identify the valid peptides identified across all 29 samples. Each non-redundant protein in the database was then iterated through and counted as identified if it met one of the following conditions. (i) More than one non-identical peptide had been identified across all samples that matched the protein. (ii) A peptide matching the protein had been identified more than once across all samples. (iii) A peptide matching the protein had been identified with spectral probability value <1E-15. After a protein had been counted as identified, all peptides matching it were removed from the pool of peptides being considered so other proteins matching these peptides would not also be counted.

## Calculation of weighted average bacterial peptides in 16S_Sample-Matched databases and correlation with relative performance of 16S_Sample-Matched and Hybrid_Sample-Matched databases

A protein sequence database is expected to generate more PSMs if it has more protein sequence data for the species present in a sample, especially those at high relative abundance which are likely to contribute a larger share of peptides to the proteome detectable by mass spectrometry. Therefore, to determine completeness of a database populated with publicly available protein sequences, a weighted mean of the number of tryptic bacterial peptides available for a search program to compare sample mass spectra against was calculated. An *in silico* tryptic digest of publicly available bacterial protein sequences was performed and the number of unique tryptic peptides longer than five amino acids was summed (43). Then, the following equation was used to calculate the weighted mean number of tryptic peptides for bacteria in each 16S_Sample-Matched database, where $P_i$ is the number of tryptic peptides available for species $i$ and $A_i$ is the relative abundance of species $i$ in the sample:

$$\sum_{i=1}^{n} \ln(1 + G_i) \times A_i$$

The relative performance of the 16S_Sample-Matched and Hybrid_Sample-Matched databases was then calculated for each sample by dividing the number of significant bacterial PSMs identified by the 16S_Sample-Matched database by the number of significant bacterial PSMs identified by the Hybrid_Sample-Matched database for that sample. The scipy (v1.3.1) "stats" package was then used to determine the Spearman's Rank-Order correlation between these values. Code in the "Taxa Abundance Correlation.ipynb" notebook was used to determine how the number of PSMs identified for different taxa changed between searches of 16S_Sample-Matched and Hybrid_Sample-Matched databases.

## Software used for initial database construction

Protein sequences from individual microbial species were downloaded from RefSeq using the DownloadFromNCBIFTPtxt.py program. Prior to database assembly, all nonhuman protein sequences were normalized with SpeciesSeqPrepper.py, then sequences from separate strains were combined into a single file with StrainCombiner.py. When a separate database tailored to each individual sample was required, a CSV file indicating which species should go into each sample's database was constructed, and code in the "Tailored DB Building.ipynb" notebook was used to assemble each 16S_Sample-Matched database. For the database, LargeDatabasePrepper.py was used to generate searchable database files approximately 300 MB in size. Code in the "Community DB Building.ipynb" notebook was used to build the 16S_Pooled database. Shotgun metagenomic sequencing databases were built by taking protein data from translated bacterial open reading frames and assembling sample-matched databases for Shotgun_Sample-Matched or pooling together all translated bacterial protein sequences for Shotgun_Pooled. Software used to build these databases is in the "Metagenomic DB Building.ipynb" Jupyter Lab notebook file. Hybrid_Sample-Matched databases were built by adding new proteins from a sample's Shotgun_Sample-Matched database to all the sequences already in its 16S_Sample-Matched database and collapsing together any proteins with identical amino acid sequences. Software used to build these databases is in the "Metagenomic Hybrid_Sample-Matched DB Building.ipynb" Jupyter Lab notebook file. 16S_Reference databases were built by identifying the reference strain for each species used to build the 16S_Sample-Matched databases. For most strains, the reference strain listed on the NCBI website was used. The only species without a listed NCBI reference strain were those for which only a single genome was available, and in these cases, the one publicly available genome was used as the reference strain. The reference strains were then placed in a CSV file, and taxa where the only available genome was to be used as the reference strain, the reference strain in the CSV file was listed as "A." Software used to build the 16S_Reference databases is in the "Reference Genome DB Building.ipynb" Jupyter Lab notebook file.

## Software used for two-step database searches

MSGF_MultipleDBs.nf and MSGF_TailoredDBs.nf were used to stage files to AWS S3, queue the searches to run on AWS EC2, and convert the search results to TSV format. For searches using a separate database for each sample, the runTailoredDB.sh script was used to initiate a search, while the runMultiDB.sh script was used when each sample was searched against a single database. Software used to create the subset 16S_Sample-Matched databases is located in the "Tailored DB Building.ipynb" Jupyter Lab notebook file. Software used to create the subset databases for Shotgun_Pooled, Shotgun_Sample-Matched, and Hybrid_Sample-Matched databases are in their respective Jupyter Lab notebook files, listed above. For the larger Global, 16S_Pooled, and Shotgun_Pooled, multiple elliot_utils.py functions were used to generate the subset databases. First, the "condenseHugeDBResults" function was used to condense the results of each sample search into a single file by comparing hits on a single spectrum and keeping the one with the highest MSGF score. "getHitsInResults" was used to identify all proteins hit in every sample regardless of significance, and "refineHugeDatabase" was used to build the subset database with those protein sequences. Results of the 16S_Pooled and Shotgun_Pooled databases were condensed into a single file with "condenseHugeDBResults" before analysis.

## Contaminant protein sequences

Proteins such as keratins and trypsins are commonly introduced into samples by processing procedures and by the researchers themselves. Therefore, the following contaminant proteins were included in all metaproteomic databases as a minimal set of peptides that should be excluded from downstream analysis while simultaneously

attempting to preserve true identifications. These proteins were: *Sus scrofa* trypsin precursor (sp|P00761|TRYP_PIG), Promega trypsin artifact 1 (Trypa1), Promega trypsin artifact 2 (Trypa2), Promega trypsin artifact 3 (Trypa3), Promega trypsin artifact 4 (Trypa4), Promega trypsin artifact 5 (Trypa5), Trypsin artifact 6 (Trypa6), *Bos taurus* trypsinogen (sp|P00760|TRYP_BOVIN), *Bos taurus* chymotrypsinogen A (CTRA_BOVIN), *Bos taurus* chymotrisinogen B (CTRB_BOVIN), *Homo sapiens* serum albumin precursor (sp|P02768|ALBU_HUMAN), *Bos taurus* serum albumin precursor (sp|P02769|ALBU_BOVIN), *Homo sapiens* keratin type II cytoskeletal 1 (K2C1_HUMAN), *Homo sapiens* keratin type II cytoskeletal 2 (K22E_HUMAN), *Homo sapiens* keratin type I cytoskeletal 9 (K1C9_HUMAN), and *Homo sapiens* keratin type I cytoskeletal 10 (K1C10_HUMAN).

## Calculation of diversity statistics from DNA sequencing data

Diversity statistics were calculated for sequencing data generated by 16S rRNA gene sequencing and shotgun metagenomic sequencing using Phyloseq (44). Code used to perform these calculations is in the "bvr01_proteomics_rarefaction.R" file. Reads from 16S rRNA gene sequencing and reads identified as bacterial DNA by MetaPhlAn2 (45) were used as the raw data. Alpha diversity was calculated by Shannon diversity, and Beta diversity between samples was visualized on multidimensional scaling plots using Bray-Curtis distance. Rarefaction curves for each sample and DNA sequencing method were generated using the "rarecurve" function in Phyloseq.

## RESULTS

### Participant characteristics

A cross-sectional case-control set of 20 vaginal samples from women with BV and 10 samples from women without BV were selected at random from the parent study of 220 women from Seattle with and without BV for analysis by 16S rRNA gene sequencing, metagenomic sequencing, and metaproteomics by LC-MS/MS (26). One of the BV-negative samples was excluded from further analysis due to the presence of a polymer or detergent that compromised the MS analysis. 41% (12/29) of study participants identified themselves as Black and 48% identified as White (14/29) (Table 1).

**TABLE 1** Characteristics of study participants

| | All participants | BV−[a] | BV+ |
|---|---|---|---|
| *N*[b] | 30[c] | 9 | 20 |
| Age range | 19–56 | 23–56 | 19–42 |
| Mean | 29.3 | 32.1 | 28.1 |
| Race/ethnicity[d] | | | |
| Black | 12 (41.4%) | 3 (33.3%) | 9 (45.0%) |
| White | 14 (48.3%) | 6 (66.7%) | 8 (40.0%) |
| Other | 2 (6.90%) | 0 (0.00%) | 2 (10.0%) |
| N/A | 1 (3.40%) | 0 (0.00%) | 1 (5.00%) |
| Nugent score | | | |
| Range | 0–10 | 0–3 | 7–10 |
| Thin homogeneous vaginal discharge | 21 (72.4%) | 3 (33.3%) | 18 (90.0%) |
| Clue cells | 19 (65.5%) | 0 (0.00%) | 19 (95.0%) |
| pH | | | |
| Range | 4.0–5.8 | 4.0–5.0 | 5.0–5.8 |
| Positive whiff test | 20 (69.0%) | 0 (0.00%) | 20 (100%) |

[a]Bacterial vaginosis was diagnosed using Amsel clinical criteria.
[b]*N* indicates the number of participants.
[c]The sample from one of the 10 BV− participants was contaminated with an unknown polymer which compromised metaproteomic analysis. Thus, this sample was excluded from further analysis and only nine BV− samples were considered.
[d]Other races included Asian/Filipino (one participant) and Native Hawaiian/Pacific Islander (one participant). One participant chose not to specify their race.

## Identified peptides differ by technical replicate

We analyzed each sample by liquid chromatography-tandem mass spectrometry (LC-MS/MS) in two separate injection replicates. In a preliminary analysis of this data, we found that 26.6% (7,001/26,285) of all unique peptides were identified only in the first run of the samples, 21.1% (5,559/26,285) were identified only in the second run, and 52.2% (13,725/26,285) were identified in both runs, reflecting the stochasticity of metaproteomic analysis (Fig. S1) (46). To maximize usable protein data, we combined both runs of each sample for all further analyses.

## Small, sample-specific databases outperform broad databases when analyzing peptides in cervicovaginal lavage samples

We sought to compare different strategies for database construction by analyzing metaproteomic data with six different types of databases: three built with publicly available sequences, two populated with proteins translated from metagenomic sequencing of the samples, and one hybrid that combined a sample-matched public sequence database and translated metagenomic database. The specific contents of each database are described in Methods (Fig. 1). Briefly, each database contained all SwissProt human protein sequences (release 2019_02) plus 16 contaminants commonly introduced during metaproteomic analysis (47). We built a Global database with all bacterial and fungal proteins available from NCBI RefSeq (RefSeq release 97). To build databases specific to the vaginal microbiome, we characterized the bacterial communities present in the samples by both 16S rRNA gene sequencing and shotgun metagenomic sequencing (Table S1; Fig. S2). In a preliminary analysis, we identified peptides from taxa present in the samples at a relative abundance as low as 0.1% according to 16S rRNA gene sequencing. Therefore, we built separate 16S_Sample-Matched databases for each sample, populated with all NCBI RefSeq proteins available for the bacteria present in the sample at >0.1% relative abundance. We then built a single 16S_Pooled database that included all bacterial proteins included in the individual 16S_Sample-Matched databases. To ensure the 16S_Pooled database better represented the broad vaginal microbiota, it also included RefSeq proteins from the common vaginal fungi *Alternia alternata, Candida albicans, Nakaseomyces glabrata, Candida tropicalis, Pichia kudravzevii,* and *Saccharomyces cerevisiae,* along with the common eukaryotic parasite *Trichomonas vaginalis*. For the translated metagenomic databases, the Shotgun_Pooled database contained the translated proteins from all bacterial open reading frames (ORFs) identified across all samples in the dataset, while the Shotgun_Sample-Matched databases only contained the translated proteins from its corresponding sample. Finally, we built a separate Hybrid_Sample-Matched database for each sample that combined the proteins in the 16S_Sample-Matched and Shotgun_Sample-Matched databases. For each database, we performed a two-step target-decoy database search as described in Methods. We evaluated the performance of the databases using the metrics time and cost for computational processing. Because protein spectral count is used to make statistical comparisons in metaproteomic analysis, we also compared the number of significant human and bacterial PSMs generated by searches of each database.

The Global database was by far the largest with 131,886,982 total protein sequences (Fig. 2A). The 16S_Pooled database was two orders of magnitude smaller, containing 1,345,203 sequences. 16S_Sample-Matched databases ranged in size from 32,054 to 401,128 sequences, depending on the microbial diversity of their matched sample. The Shotgun_Pooled database contained 443,291 sequences, while the Shotgun_Sample-Matched databases ranged from 21,736 to 111,694 sequences. The Hybrid_Sample-Matched databases contained between 32,587 and 474,108 proteins. Because the Global database was so large and computationally expensive to search, we randomly selected a subset of six samples (two BV negative samples and four BV positive samples to maintain the proportion of the initial sample set) to search against it and compare to the other database types. The computational power required to perform a two-step search against

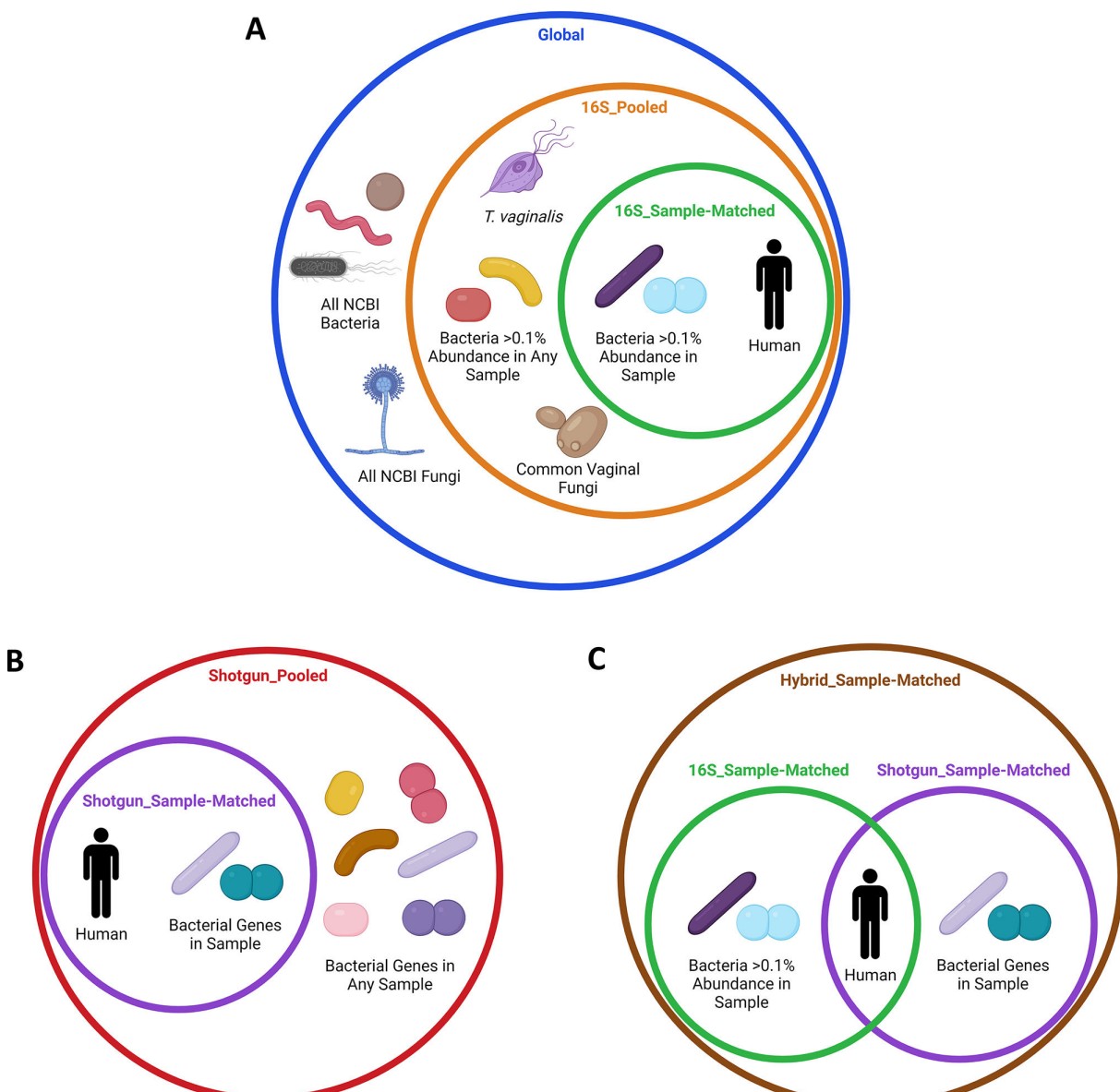

**FIG 1** Contents of the public sequence and translated metagenomic sequencing databases. Illustration of proteins included in the tested databases. (A) Proteins used to make the public sequence databases. (B) Proteins used to make the translated shotgun metagenomic databases. (C) Proteins used to make the hybrid public sequence/translated shotgun metagenomic databases. Created with BioRender.com.

each database type was proportional to their initial size, with the Global database requiring approximately 100-fold more central processing unit (CPU) hours than the 16S_Pooled database (Fig. 2B). Based on standard pricing for Amazon Web Services cloud computing at time of writing these CPU requirements equate to an average of $155.36 to perform a two-step search on one sample using the Global database, $1.63 with the 16S_Pooled database, $0.31 with a 16S_Sample-Matched database, $0.65 with the Shotgun_Pooled database, $0.20 with a Shotgun_Sample-Matched database, or $0.32 with a Hybrid_Sample-Matched database (Fig. 2C).

We analyzed the number of significant fungal PSMs generated by searches of Global and 16S_Pooled databases, as defined by a $q$ value <0.01, since these were the only databases that included fungal proteins. Searches of the subset of six samples for both databases identified a very small number of significant fungal PSMs, 23 on average for Global and 1 for 16S_Pooled (Fig. S3). Because the number of fungal proteins in the

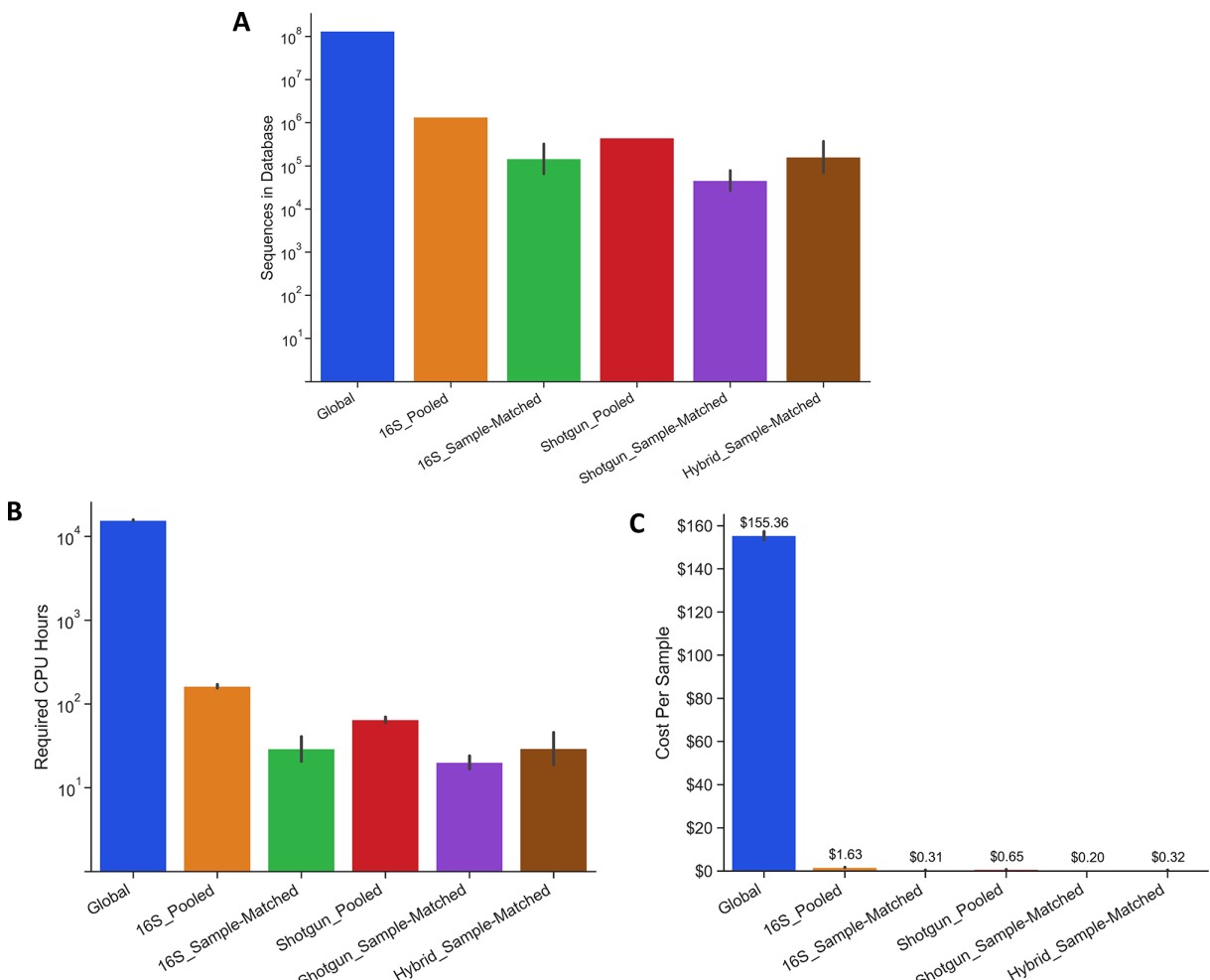

**FIG 2** Comparison of database types: size, computational power, and cost. (A) Comparison of the number of protein sequences in each database type. Lines show standard deviation in cases where a separate database was built for each sample. (B) Total CPU hours required to perform a two-step search using each database type with the MS-GF+ search program. Lines show standard deviation to search the subset of six samples. (C) Cost to perform a two-step search using each database type. Numbers represent the average cost to run the subset of six samples. Lines show standard deviation to search the subset of six samples. Graphs show mean ± standard deviation.

samples appeared to be so low, we focused the rest of our analysis on human and bacterial PSMs.

Database size was a major factor in the number of significant human and bacterial PSMs generated, as the relatively small 16S_Sample-Matched databases generated the most significant human and bacterial PSMs of the three public sequence databases (Fig. 3A and B). This pattern held for all 29 vaginal samples (Table S2). We performed a pairwise comparison of each database type (excluding the Global database) to determine which databases generated more significant human and bacterial PSMs (Fig. S4A and B) and calculated relative identification rates for each database to visualize these comparisons (Fig. 4A and B). Shotgun_Sample-Matched databases identified significantly more human and bacterial PSMs than the Shotgun_Pooled database, while the 16S_Sample-Matched databases similarly outperformed Global and 16S_Pooled. The relative performance of public sequence versus translated metagenomic databases varied by PSM type. The smaller Shotgun_Sample-Matched databases generated significantly more human PSMs than all other databases, but 16S_Sample-Matched databases significantly outperformed them in terms of bacterial PSMs. Hybrid_Sample-Matched database searches combined novel PSMs from both database types, identifying

as many or more bacterial PSMs than the 16S_Sample-Matched databases in 24 of 29 samples (83%), and only slightly fewer human PSMs (Table S2). We also compared the average percent of bacterial PSMs that only matched one or more proteins from a single genus or species (Table S3). Public sequence databases outperformed the translated metagenomic databases when assigning PSMs at the genus level; however, all databases performed similarly at the species level, assigning between roughly 40% and 50% of bacterial PSMs to a single species.

To determine whether generating additional significant PSMs translated into more biological insights, we compared the number of differentially abundant Gene Ontology (GO) terms associated with proteins between BV+ and BV− samples (48, 49). We used EggNOG-Mapper to collect GO terms for each protein (41, 42). Searches that generated a greater number of significant PSMs also identified more functions that were significantly differentially abundant by BV status (Table 2). Results from Shotgun_Sample-Matched searches, which identified the most human PSMs in all samples, led to the most significantly differentially abundant human functions. Similarly, results from the Hybrid_Sample-Matched database searches led to the most significant differentially abundant bacterial functions.

## An extremely broad protein database is prone to produce false-positive identifications, while a focused database containing taxa expected to be in the community is more accurate

The Global and 16S_Pooled databases contained sequences from many microbial taxa that were not present in a given sample, so we evaluated whether this impacted the accuracy of their searches. We analyzed the significant bacterial PSMs identified by the Global and 16S_Pooled databases and determined what percent exclusively matched proteins from taxa at <0.1% abundance (based on 16S rRNA gene sequencing) in the sample, likely representing false positives. Of the bacterial PSMs detected by the 16S_Pooled database search, less than 0.5% on average exclusively matched genera below 0.1% relative abundance in the sample (Fig. 5). In contrast, an average of 20% of significant bacterial PSMs identified by searches of the global database only matched genera that were not likely present in the sample. *Mycolicibacterium, Fluoribacter*, and *Legionella* were some of the genera most often falsely attributed to bacterial spectra by searches of the Global database (data available at https://figshare.com/s/f3e0ecdddf34189e90b5).

## Shotgun_Single databases outperform the Shotgun_Pooled database because their smaller size reduces the threshold of significance for spectrum identifications

Shotgun_Sample-Matched database searches identified more bacterial PSMs than the Shotgun_Pooled database in significantly more samples (P < 0.0001). This result could either indicate that the additional sequences in the Shotgun_Pooled database do not lead to novel spectrum identifications or that the smaller size of the Shotgun_Sample-Matched databases requires a lower statistical threshold for individual PSMs. To determine whether the pooled approach identified any new spectra, we investigated bacterial spectra given a significant identification by only one of the Shotgun databases, or both (Fig. 6). All samples had at least one bacterial spectrum identified only by the Shotgun_Pooled database search, showing that additional protein sequences resulted in identifications of novel spectra. We also evaluated spectra successfully identified by Shotgun_Sample-Matched database searches but missed by those of the Shotgun_Pooled database. For 92% of these spectra, MS-GF+ assigned them the same peptide sequence regardless of the database searched, but with a q value greater than 0.01 for the Shotgun_Pooled searches. This indicates that the larger size of the Shotgun_Pooled database pushed these spectra below the threshold for significance and reduced the overall search performance.

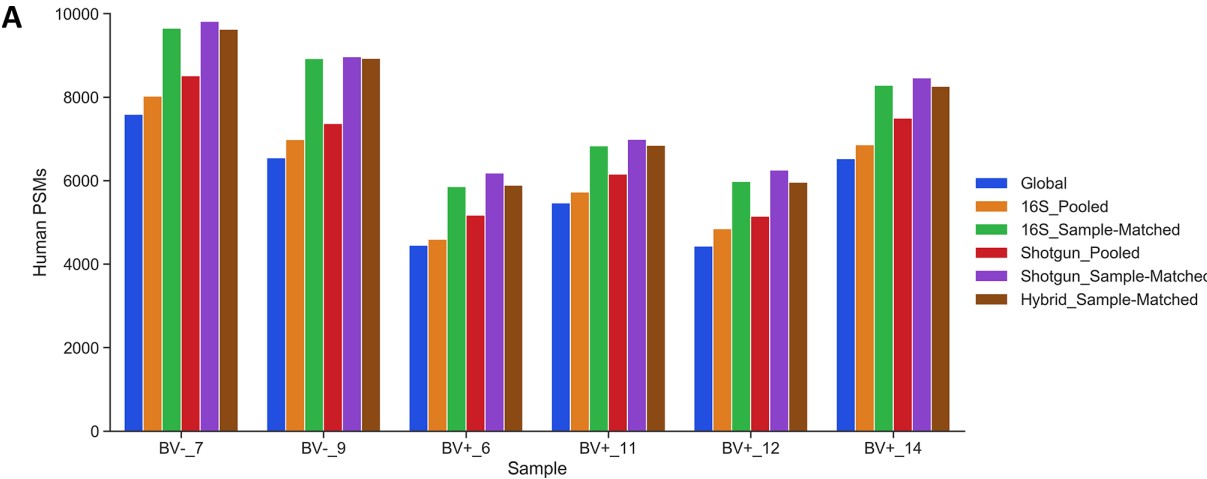

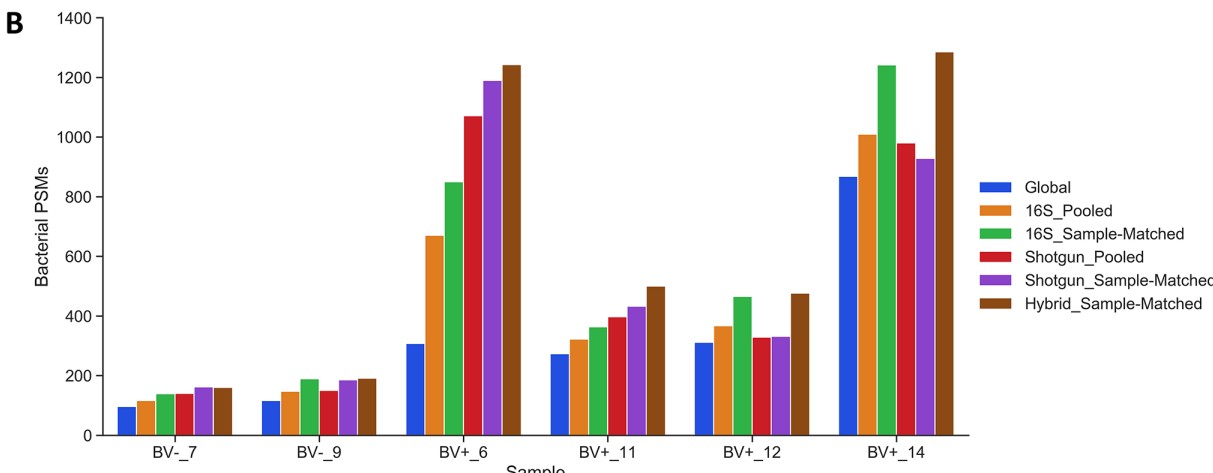

**FIG 3** Comparison of significant human or bacterial PSMs generated by six database types. Number of significant human (A) and bacterial (B) PSMs generated by searching the subset of six samples using each database type.

## Minimizing the number of bacterial proteins in a public sequence database slightly increases significant human PSMs but decreases identified bacterial spectra

Counterintuitively, one of the primary determinants of the number of significant human PSMs generated by a database search may be the number of bacterial proteins in the

**TABLE 2** Differences in significantly differentially abundant functional annotations between BV− and BV+ samples by database type[a]

| Database type | Significantly different human functions | Significantly different bacterial functions |
| --- | --- | --- |
| 16S_Pooled | 1,713 | 386 |
| 16S_Sample-Matched | 1,881 | 460 |
| Shotgun_Pooled | 1,710 | 377 |
| Shotgun_Sample-Matched | 1,935 | 411 |
| Hybrid_Sample-Matched | 1,903 | 479 |

[a]Number of Gene Ontology (GO) numbers that were significantly differentially abundant between BV− and BV+ samples based on the results of different database searches. Test for significance was carried out by Mann-Whitney U test for individual proteins/functional annotations using a significance of $P < 0.01$.

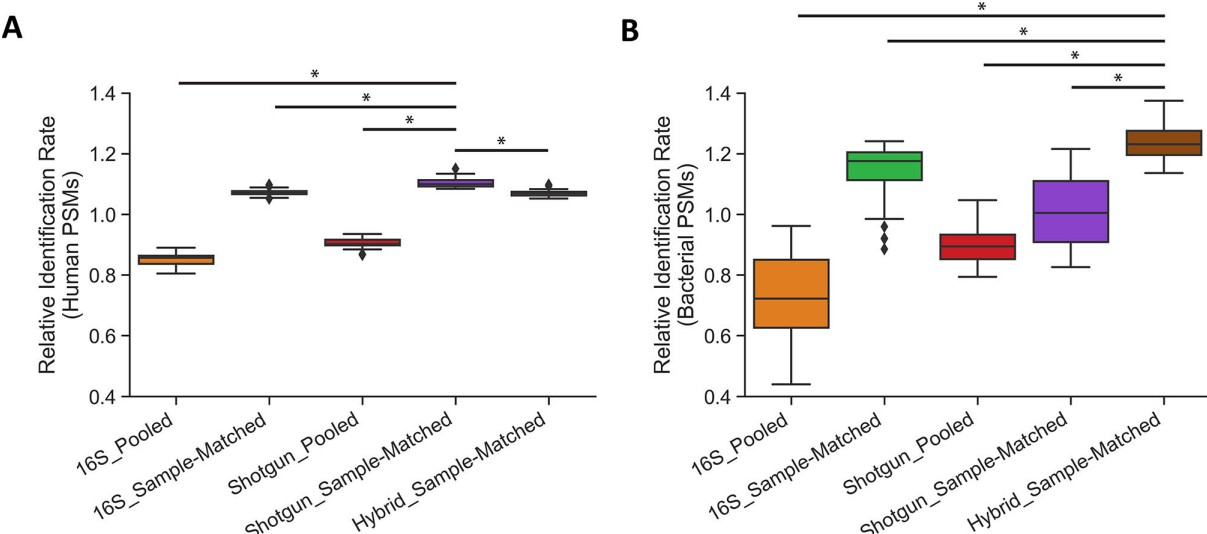

**FIG 4** Relative identification rates between database types. Relative identification rate for each sample database search was calculated by taking the number of significant human (A) or bacterial (B) PSMs identified in a particular sample using the listed database, then dividing this number by the average significant PSMs of the same type identified in that sample across all five database searches. Bars above the data represent pair-wise comparisons of significant PSMs generated that were significantly different according to a Wilcoxon signed-rank tests ($P < 0.01$). These bars are only depicted for the database that performed best for the given PSM type. Results of all pair-wise database comparisons are shown in Fig. S4.

database, as many more bacterial proteins had to be included in the database to account for the heterogeneity of the bacterial proteome. Therefore, we tested whether minimizing the number of bacterial proteins in a database would increase the number of significant human PSMs it generated. For each sample, we built a 16S_Reference database that included proteins from each bacterial species present in a sample at >0.1% relative abundance but only from the reference strain of that species as listed by NCBI (data available at https://figshare.com/s/1d51e388676e695c4cf5) (37). Searches of these pared-down databases on average resulted in approximately 2% more significant human spectrum identifications than the 16S_Sample-Matched databases, but searches of the

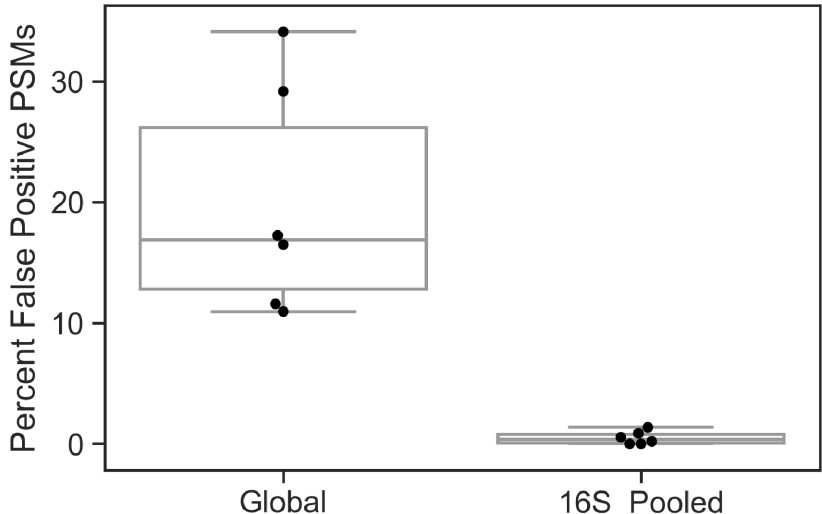

**FIG 5** Presumptive false positive hits in Global and 16S_Pooled database searches. Each data point represents the percent of PSMs in a given sample in which MS-GF+ matched to protein(s) belonging only to bacteria known to not be present in the sample, as determined by a relative abundance <0.1%. Results are shown from the subset of six samples searched using the Global database.

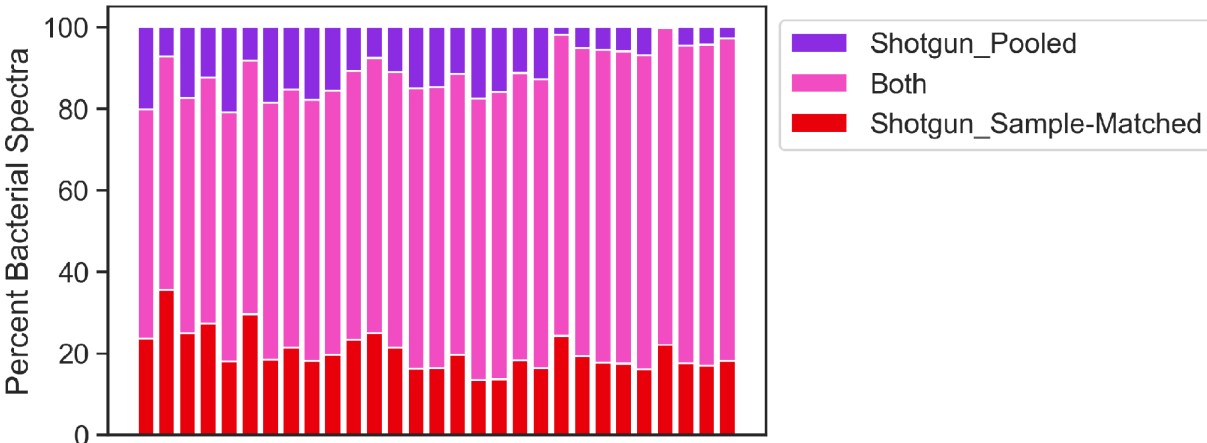

**FIG 6** Overlap of spectra identified in searches of Shotgun_Pooled and Shotgun_Sample-Matched databases. Percent of bacterial spectra in each sample identified only by searching the Shotgun_Sample-Matched database (red), Shotgun_Pooled database (purple), or identified in both database searches (pink).

16S_Sample-Matched databases resulted in approximately 19% more significant bacterial spectrum identifications, on average (Fig. 7). There was a large range in the fold-change difference in bacterial PSMs between the two databases, however, and in one sample the 16S_Reference search identified fewer than 40% of bacterial PSMs identified by the 16S_Sample-Matched search.

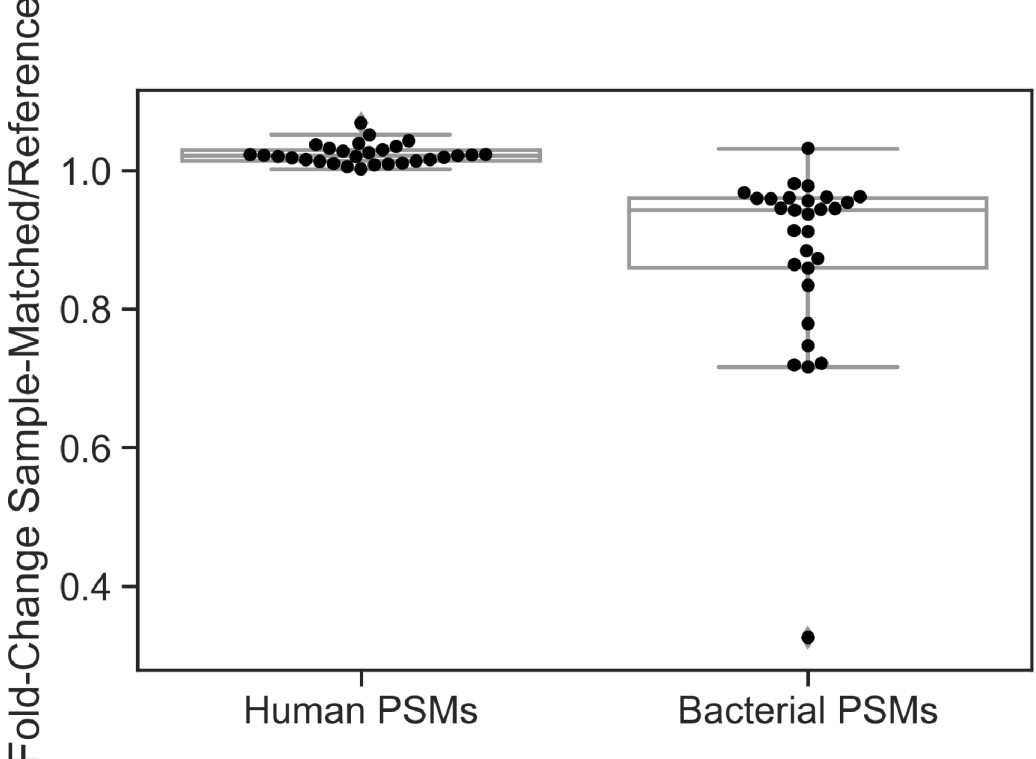

**FIG 7** Fold-change in statistically significant human and bacterial PSMs identified by the 16S_Sample-Matched database searches compared to searches of 16S_Reference databases. Each data point represents the ratio of significant PSMs identified by the 16S_Sample-Matched database compared to the 16S_Reference database in a sample.

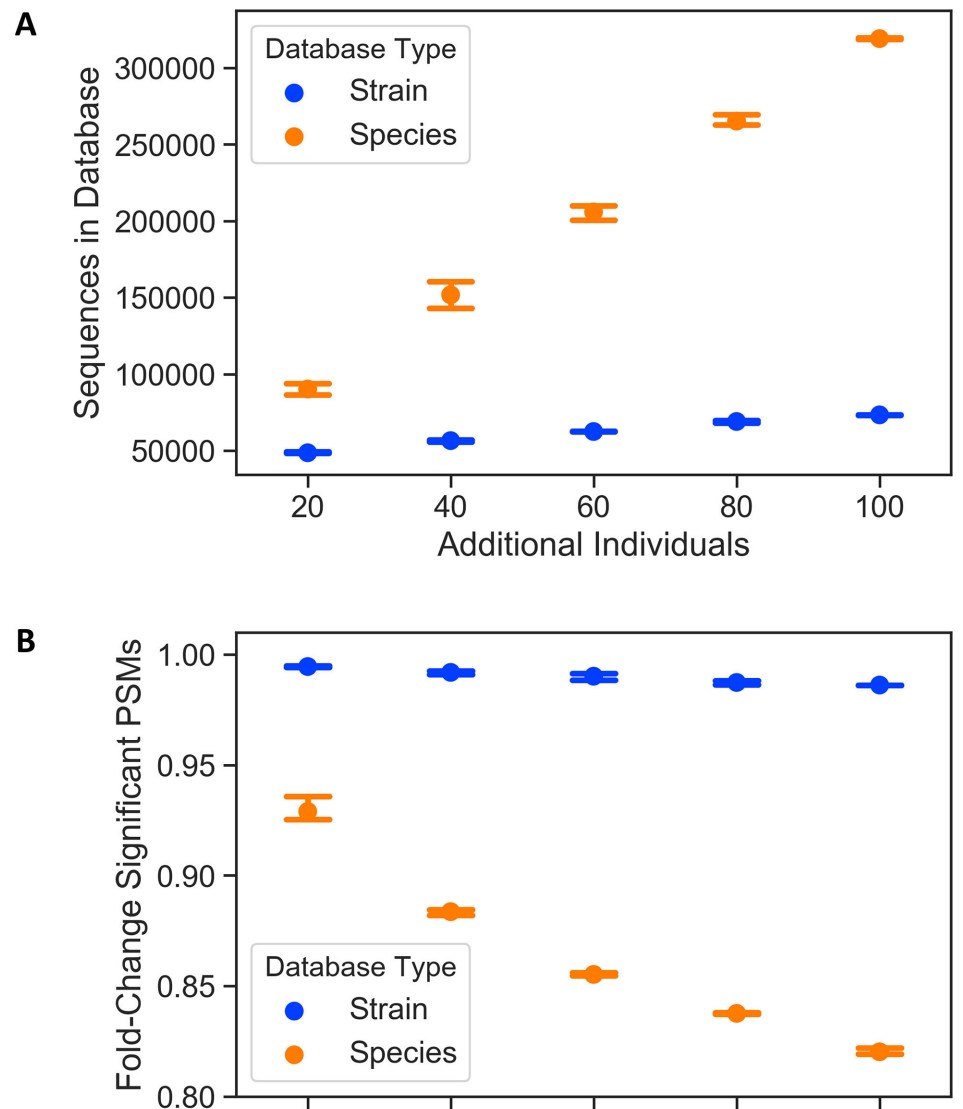

**FIG 8** Effect of additional protein sequences from strains or species on database search performance. Dots represent mean values, and bars represent standard error. Databases were built by randomly adding a set number of individuals from a list of 103 strains of *L. crispatus* or 103 different bacterial species taken from the human microbiome sequencing project. (A) Total number of sequences in the database. Data represents three randomly generated databases at each size. (B) Fold-change in number of significant PSMs generated as compared to the baseline database. The value for each individual database was the mean fold-change for all six samples searched. Data represents three randomly generated databases at each size.

## Adding proteins to a database from additional strains of a species has a less negative effect than adding proteins from additional species due to greater protein heterogeneity

While our comparisons of the Global, 16S_Pooled, and 16S_Sample-Matched databases showed that database size is a major factor in search performance, our tests of the 16S_Reference databases indicated that including proteins from many strains of the species present in a sample also increased significant bacterial PSMs generated by a search. To test how adding additional strains or species to a database impacts search performance, we constructed a set of test databases by iteratively adding protein sequences from additional random bacterial species from the Human Microbiome

Project (50) or strains of *Lactobacillus crispatus* to a baseline database. Due to overlapping protein sequences of the different *L. crispatus* strains, databases with additional species increased in size much faster compared to the databases with additional strains (Fig. 8A). We performed one-step searches using these databases on six of the proteomic samples: three low-diversity samples dominated by *L. crispatus* (>50% relative abundance) and three high-diversity samples not dominated by *L. crispatus* (<50% relative abundance). The number of significant PSMs decreased much faster for the databases with additional species than the databases with additional strains, reaching a ~17% reduction when protein sequences from 100 different species were added (Fig. 8B).

## Public sequence databases identify many peptides from highly abundant taxa while translated shotgun sequencing databases excel at identifying peptides from uncultivated species

In most samples, searches of both the 16S_Sample-Matched and Shotgun_Sample-Matched databases identified many bacterial spectra that were missed by the other database type (Fig. S5). We investigated the taxonomic identifiers associated with these spectra to determine whether the different database types were better at identifying peptides from certain groups of bacteria (Table S4). Many of the spectra that were exclusively identified by the 16S_Sample-Matched database were associated with taxa that were in high abundance in the samples, including *Gardnerella, Prevotella, Fannyhessea,* and *Lactobacillus*. The majority of these spectra were associated with *Gardnerella*, which is a well-sequenced, highly diverse genus that is frequently the dominant community member in BV. This indicates that shotgun metagenomic sequencing can miss genes from major bacterial community members that are well represented in public sequence databases, likely as a result of low inherent sampling depth in metagenomic sequencing. Of those spectra exclusively identified by the Shotgun_Sample-Matched database searches, the largest number were associated with the taxonomic identifiers "Unknown" or "Terrabacteria." Therefore, many of the bacterial genes identified by shotgun metagenomic sequencing could not be confidently resolved to a single group of bacteria. However, more than 300 spectra exclusively identified by the Shotgun_Sample-Matched database were assigned the taxa label "Clostridiales bacterium KA00274." This bacterium is closely related to BV-associated bacterium 2 (BVAB2), a bacterium with no currently published genomes. These results demonstrate a situation where shotgun metagenomic sequencing identified bacterial peptides present in a sample but absent from public repositories.

## Results from optimized database searches identify a large number of unique proteins while remaining in line with past metaproteomic studies of the vaginal microbiome

Past metaproteomic studies of vaginal samples have primarily used the Uniprot/TrEMBL database as a reference for their bacterial proteins (Table S5). However, we found that a more tailored approach outperformed large, very broad databases. Our sample-matched databases identified a large number of unique proteins compared to other metaproteomic studies of vaginal samples, relative to the number of samples analyzed (Table S5). Shotgun_Sample-Matched database searches identified the most unique human proteins at 1,182, while Hybrid_Sample-Matched identified the most bacterial proteins at 1,418. Although it may be valuable to identify more proteins, it is important that data remains accurate. To verify the accuracy of our optimized database searches, we identified human proteins which past metaproteomic studies had found were differentially abundant depending on BV status. We then tested whether there were similar differences in our results. Because the Hybrid_Sample-Matched databases struck the best balance between human and bacterial data, we analyzed the results of these database searches. The total number of human PSMs we identified in a sample also varied depending on BV status (data not shown), so we normalized the spectral count

of each protein to the number of human PSMs identified in that sample, and compared the relative abundance of each protein between samples. Twenty-three human proteins had been identified as significantly differentially abundant in at least one of the studies analyzed (51–55). In our data, we found significant differences in agreement with past studies for nine of these proteins (Table 3). Two proteins in our data with significantly different abundances did not align with past reports.

## Shotgun metagenomic sequencing may provide less benefit to metaproteomic analysis of samples with a high abundance of well-sequenced species

Although Hybrid_Sample-Matched database searches identified the most significant bacterial PSMs, in 17 of the 29 samples, the 16S_Sample-Matched databases identified only 5% fewer bacterial PSMs. Performing additional shotgun metagenomic sequencing adds cost and complexity to analysis, so for future investigations, it would be useful to predict whether creating Hybrid_Sample-Matched databases would provide substantially more metaproteomic data. We hypothesized that having more protein sequence data for the bacteria in a sample would correlate with better relative performance of 16S_Sample-Matched databases. To test this hypothesis, we summed the number of tryptic peptide sequences available for the bacterial species present in each sample, weighted to their relative abundance. We then compared this number to the ratio of significant bacterial PSMs identified by the 16S_Sample-Matched database compared to its equivalent Hybrid_Sample-Matched database (Fig. 9). A Spearman's rank-order correlation test found a statistically significant correlation between an increasing number of tryptic peptide sequences available to search and the relative performance of the public sequence-only databases, showing that the benefits of metagenomic sequencing decrease as the amount of sequence information available for organisms increase. We also investigated what organisms had the largest increase in the total number of significant PSMs identified across all samples by utilizing a Hybrid_Sample-Matched database. Some of the largest were Candidatus *Lachnocurva* (29 additional PSMs), *Megasphaera hutchinsoni* (57 additional PSMs), and BVAB2 (378 additional PSMs). All three of these species had very little protein sequence data available at the time of analysis (only one published genome for both Candidatus *Lachnocurva* and *M. hutchinsoni* and no available genomes for BVAB2), and high relative abundance in at least one sample (72.3%, 10.7%, and 10.4%, respectively). Conversely, *L. crispatus* and *L. iners* both have a large number of published genomes and had >80% relative abundance in multiple samples, but using searches of the Hybrid_Sample-Matched databases, we identified 2 fewer total PSMs for *L. crispatus* and 13 fewer total PSMs for *L. iners*.

## DISCUSSION

Metaproteomic methods have great potential to illuminate microbial physiology and host-microbe interactions in microbial communities, but few studies have systematically examined the impact of protein database composition on metaproteomic results. There are dangers both in including too many proteins in a database and too few. If proteins present in a sample are not included in the database, their spectra can be misidentified and misattributed (19). Conversely, excess proteins in a database increase FDR and raise the threshold for a PSM to be significant (56). Although a database should theoretically include as many proteins as possible to best capture the diversity of a sample, in practice, increasing statistical stringency drives PSMs below the threshold for significance, reducing the useable data generated. Many approaches have been proposed to balance these forces including employing a two-step search strategy, *de novo* peptide sequence identification, translating sequenced mRNA from study samples, populating databases with short "metapeptides," and combining public sequences with translated genes from metagenomic sequencing (40, 57–61).

In this study, we tested multiple database types to evaluate the relative performance of different database construction strategies on metaproteomics results from vaginal samples. We found that the number of bacterial protein sequences included in a

**TABLE 3** Comparison of differentially abundant human proteins in BV identified by past studies with results of Hybrid_Sample-Matched database searches[a]

| Protein | This study | | Past studies | |
|---|---|---|---|---|
| | *P*-value | Higher/lower in BV | Higher/lower in BV | Reference |
| Elafin | >0.1 | ↓ | ↓ | Stock et al. (51) |
| Muc5B | <0.05 | ↑ | ↑ | Borgdorff et al. (52) |
| Muc5AC | <0.05 | ↑ | ↑ | Borgdorff et al. (52) |
| Calprotectin | >0.1 | ↓ | ↑ | Borgdorff et al. (52) |
| Complement factor 3 | >0.1 | ↑ | ↑ | Borgdorff et al. (52) |
| Migration inhibitory factor | >0.1 | ↑ | ↑ | Borgdorff et al. (52) |
| Cystatin A | <0.01 | | ↓ | Borgdorff et al. (52) |
| | | ↓ | ↓ | Ferreira et al. (54) |
| Lysozyme C | >0.1 | ↓ | ↓ | Borgdorff et al. (52) |
| Serine protease inhibitor kazal type 5 | <0.1 | ↓ | ↓ | Borgdorff et al. (52) |
| Involucrin | <0.01 | | ↑ | Zevin et al. (53) |
| | | ↓ | ↓ | Ferreira et al. (54) |
| Cornifin-A | <0.05 | ↓ | ↑ | Zevin et al. (53) |
| Cathepsin G | >0.1 | ↑ | ↑ | Ferreira et al. (54) |
| Neutrophil elastase | >0.1 | ↑ | ↑ | Ferreira et al. (54) |
| Neutrophil defensin 1 | >0.1 | ↑ | ↑ | Ferreira et al. (54) |
| Leukocyte elastase inhibitor | <0.01 | ↓ | ↓ | Ferreira et al. (54) |
| Histone H4 | >0.1 | ↓ | ↓ | Ferreira et al. (54) |
| SPR3 | <0.01 | ↓ | ↓ | Ferreira et al. (54) |
| Cornifin-B | <0.01 | ↓ | ↓ | Ferreira et al. (54) |
| SPR2A | >0.1 | ↓ | ↓ | Ferreira et al. (54) |
| Repetin | >0.1 | ↑ | ↓ | Bradley et al. (55) |
| Keratin 6A | >0.1 | ↑ | ↓ | Bradley et al. (55) |
| Keratin 16 | <0.01 | ↑ | ↓ | Bradley et al. (55) |
| Suprabasin | <0.01 | ↓ | ↓ | Bradley et al. (55) |

[a]Human proteins were found to be differentially abundant in BV in past metaproteomic studies compared with results from this investigation. Mann-Whitney U tests were performed on theresults of Hybrid_Sample-Matched database searches and significance levels are shown. Each cell in the table specifies whether the protein had a higher (orange up-arrow) or lower (blue down-arrow) abundance in BV+ samples compared to BV− samples.

database has a large effect on the number of human PSMs generated by a search. Higher numbers of bacterial proteins in a database resulted in higher statistical thresholds, driving many human PSMs below the cutoff for significance. This will be a challenge for analyzing samples from many body sites, as the heterogeneity present in the bacterial proteome will often be greater than that of the human proteome. We also found that a database tailored to the vaginal microbial community (16S_Pooled) generated a substantial number of PSMs with few obvious false positive identifications while a maximally broad database (Global) generated fewer PSMs with lower accuracy, and at a much higher cost per sample searched. This result agrees with prior investigations which show large protein databases underperform more focused databases (18, 20–22, 62, 63). We also built 16S_Reference databases to test how limiting the number of bacterial proteins in a database, even from the species known to be present in the sample, affects search results. Surprisingly, these minimal databases still generated approximately 88% as many significant bacterial PSMs as the larger 16S_Sample-Matched databases, though with a much lower percentage in some samples. The 16S_Reference database also generated on average 2% more human PSMs, likely because of the reduced size relative to the 16S_Sample-Matched databases.

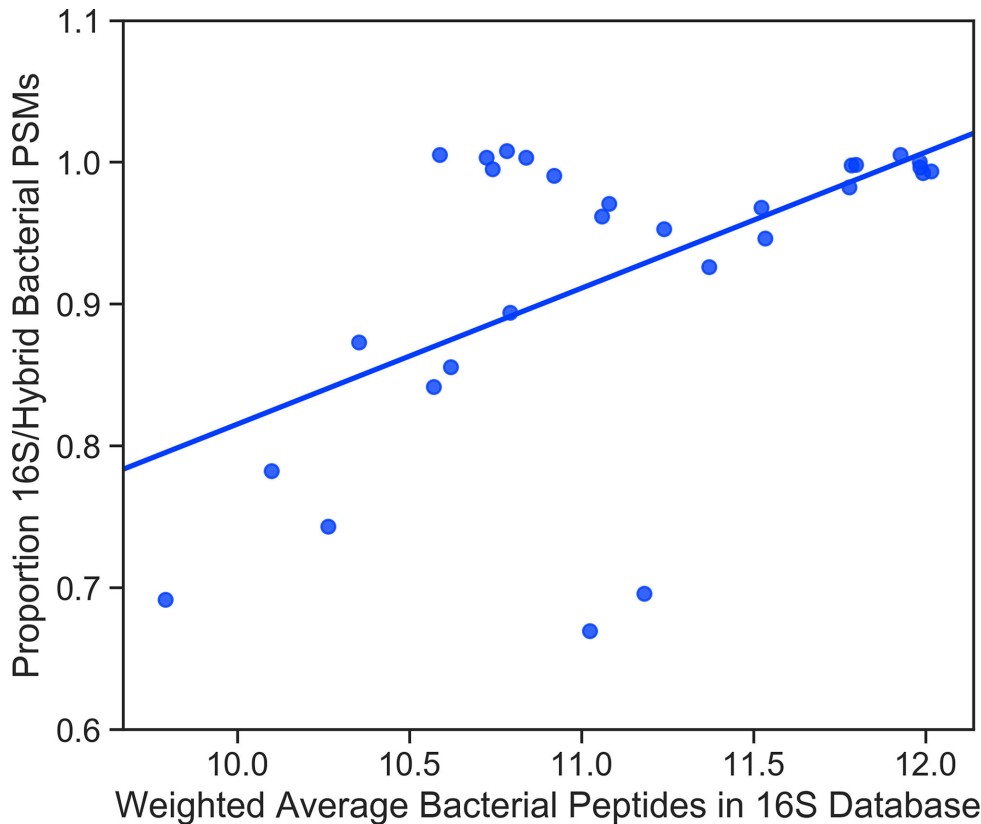

**FIG 9** Correlation between increasing public protein sequence data available for species in a sample and performance of 16S_Sample-Matched databases compared to Hybrid_Sample-Matched databases. An *in silico* tryptic digest was performed on all publicly available protein sequences for bacteria present in the samples to determine how much protein sequence data was available for searching. The amount of sequence data in each 16S_Sample-Matched database was then calculated by log-transforming the number of tryptic peptides available for each species in the sample, weighted to its relative abundance, and summed across all species. The correlation between this weighted average number of tryptic bacterial peptides present in a 16S_Sample-Matched database is shown against the ratio of significant bacterial PSMs for that sample identified by 16S_Sample-Matched over Hybrid_Sample-Matched database searches. Line of best fit as calculated by linear regression is shown. Spearman's rank-order correlation: $\rho$ (46) = 0.42, $P < 0.05$.

Proteins identified as significantly differentially abundant in searches of the Hybrid_Sample-Matched databases largely agreed with past metaproteomic studies of BV. The exceptions were Cornifin A, which our data indicated had a significantly lower abundance in BV, and Keratin 16, which had a significantly higher abundance in BV. Cornifin A is a component of the cornified envelope of keratinocytes, and the trend in our data matched other cornified envelope proteins that were significantly less abundant in BV. Compared to other metaproteomic studies of the vagina, searches of our optimized databases also identified a relatively large number of unique human and bacterial proteins, showing the added value of optimized databases for metaproteomics. An outlier in this comparison was the investigation by Nunn et al., which identified a large number of proteins (3,334 human and 1,092 bacterial) from only four samples (64). Their study analyzed vaginal mucus collected by Softcup, whereas most published metaproteomic studies of the vagina, including ours, used CVL. Vaginal mucus collected by Softcup may therefore be a richer sample type for future metaproteomic studies.

Past studies have generally found databases built using genes from metagenomic sequencing outperform public sequence databases, likely because they better represent the proteomic diversity of the sample (21, 22, 62, 63). In contrast, our 16S_Sample-Matched databases generated more bacterial PSMs than the Shotgun_Pooled database in 26 of the 29 samples, and the Shotgun_Sample-Matched databases in all but six

samples. Some unique features of the vaginal niche may be responsible for these results. Past studies have primarily focused on the gut microbiome where bacterial diversity is often an order of magnitude higher (25, 65, 66). There are also many more published genomes available for the species of bacteria common in the vagina compared to those in the gut. Therefore, the relatively large number of publicly available genomes for species of vaginal bacteria may represent deeper sequencing than shotgun metagenomic sequencing of the actual samples. Additionally, because human proteins make up a greater proportion of total proteins in vaginal samples compared to the more bacteria-dense gut, statistical constraints from large numbers of bacterial proteins in a database will have a larger impact on the total number of PSMs generated. These features of the vaginal microbiome manifest in the results of our Hybrid_Sample-Matched databases. These databases combined the deep sequencing represented in public repositories with the heterogeneity identified by shotgun metagenomic sequencing of the samples while remaining small enough to have a minor impact on the number of human PSMs generated.

Our study provides additional evidence that a combined public sequence/metagenomic approach to database construction leads to additional significant bacterial PSMs (23, 61), but shotgun metagenomic sequencing adds additional cost and complexity to a metaproteomic study, in some cases without generating more PSMs. We found a correlation between the number of tryptic bacterial peptides available in public repositories and the relative performance of a protein database that only included publicly available sequences. Additionally, we found that PSMs identified by leveraging shotgun metagenomic sequencing in Hybrid_Sample-Matched databases often came from taxa with relative abundances ranging from ~10–70% in the samples but with little to no publicly available protein sequence data.

The results of our investigation lead to some general suggestions for constructing protein databases for metaproteomic analysis to fit the goals and limitations of a study (Fig. 10). First, if the primary goal of a study is to investigate the human proteome, it is important to note that including more bacterial proteins in the database will drive down the number of human PSMs generated by a search, even when using a two-step search strategy. Including proteins only from the reference genomes of the bacteria present in the sample will likely still provide a substantial amount of data, though at the expense that some bacterial protein diversity will be lost. Second, if bacterial proteins are also of interest in the study, performing 16S rRNA gene sequencing to profile the bacterial community of each sample and create sample-matched databases will generate more bacterial PSMs than a pooled approach, though PSMs from a community database are likely still useful. Third, databases which only include publicly available protein sequences may be adequate for analyzing samples composed of bacteria with many published genomes. However, if species with little available protein sequence data make up a substantial proportion of the community, an analysis will likely be improved by performing metagenomic sequencing.

While these guidelines are likely to help increase human and bacterial PSMs in other studies, we only tested them on samples from the vaginal microbiome. Database composition may have different effects on metaproteomic studies of other body sites or on purely microbial samples. Additionally, although many search programs are available for metaproteomic studies (17, 67), we only used the MS-GF+ search program on our samples. MS-GF+ is primarily designed to search smaller protein databases, which may have negatively impacted the performance of the large Global database. However, MS-GF+ outperforms other frequently used search programs such as Mascot, SEQUEST, and MS-Align+ (68, 69), hence our selection of this search program for our study.

We did not investigate every variable involved in protein database construction. For example, while the relatively large number of bacterial proteins had an impact on the number of significant human PSMs generated by a database search, including all human proteins in each database increases database size and may drive down the number of significant bacterial PSMs generated by a search. Future studies could

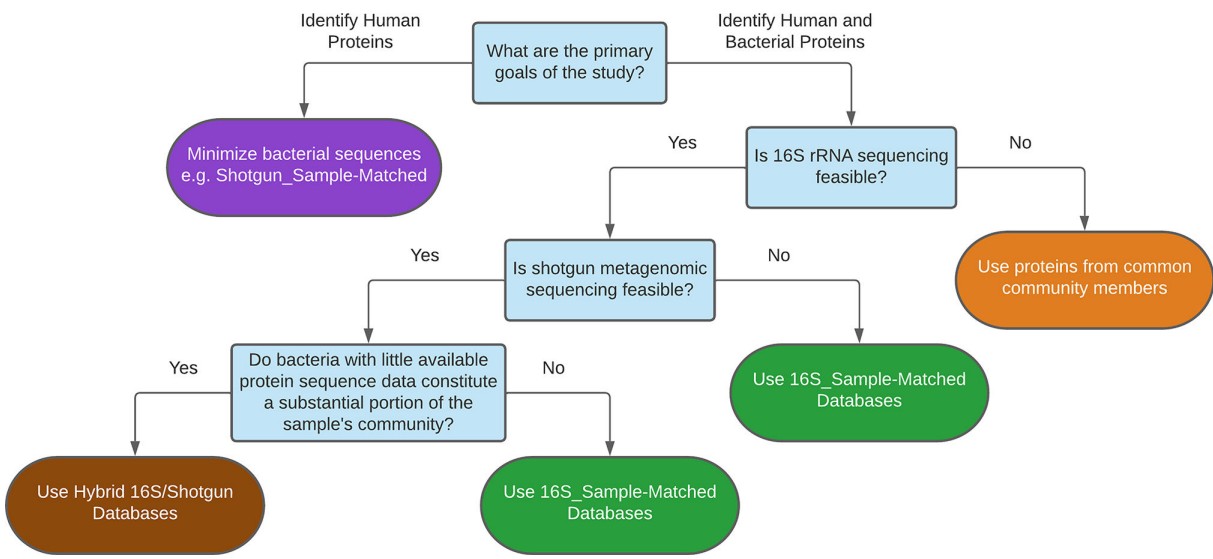

**FIG 10** Decision tree for selection of a metaproteomic database to achieve study goals with available resources. Created using lucid.app.

investigate whether filtering human proteins out of a database based on body site or by referencing RNA-sequencing data could lead to more useful data on the microbial metaproteome (13). Additionally, optimal construction of public sequence databases could be more thoroughly explored, especially regarding what species to include in the database. Our data suggested that databases tailored to the sample based on 16S rRNA gene sequencing data outperform other public sequence databases, and in many cases, databases assembled from shotgun metagenomic sequencing. We tested databases that included all species present in the sample at >0.1% abundance, however, it is possible that databases built with different cutoff thresholds could increase performance. We chose this cutoff because on average 2.7% of significant bacterial PSMs nonexclusively matched at least one taxa with relative abundance between 1% and 0.1% (data not shown). However, because the majority of identifiable spectra will likely come from species at higher abundance, it is possible that focusing on these taxa (e.g., >1% abundance) will increase overall database performance at the expense of functional information for minority community members.

   This study provides guidance on database construction for future metaproteomic studies. Our findings support past investigations suggesting that small, focused protein databases increase significant PSMs generated in metaproteomic analysis, leading to more biological insights. We also show that it is important to consider the specific niche under investigation when building a protein database as microbial diversity and availability of genomic information will impact the performance of different approaches.

## ACKNOWLEDGMENTS

This work was supported by National Institutes of Health grant R01AI061628. Portions of this research were supported by the NIH National Institute of General Medical Sciences (GM103493) and the W.R. Wiley Environmental Molecular Science Laboratory (a national scientific user facility sponsored by the U.S. Department of Energy's Office of Biological and Environmental Research and located at Pacific Northwest National Laboratory). Pacific Northwest National Laboratory is operated by Battelle Memorial Institute for the U.S. Department of Energy under contract DE-AC05-76RLO-1830.

   Some figures were made with BioRender.com.

## AUTHOR AFFILIATIONS

[1]Fred Hutchinson Cancer Research Center, Seattle, Washington, DC, USA

²University of Washington, Seattle, Washington, DC, USA
³Pacific Northwest National Laboratory, Richland, Washington, DC, USA

## AUTHOR ORCIDs

Elliot M. Lee  http://orcid.org/0000-0001-8323-2795
Sujatha Srinivasan  http://orcid.org/0000-0002-7267-3859
David N. Fredricks  http://orcid.org/0000-0003-2735-6800

## FUNDING

| Funder | Grant(s) | Author(s) |
|---|---|---|
| HHS | National Institutes of Health (NIH) | R01AI061628 | Elliot M. Lee |
| | | Sujatha Srinivasan |
| | | Tina L. Fiedler |
| | | Sean C. Proll |
| | | David N. Fredricks |
| HHS | NIH | National Institute of General Medical Sciences (NIGMS) | GM103493 | Samuel O. Purvine |
| | | Owen P. Leiser |
| | | Brooke L. Deatherage Kaiser |

## AUTHOR CONTRIBUTIONS

Elliot M. Lee, Conceptualization, Data curation, Formal analysis, Investigation, Methodology, Project administration, Software, Visualization, Writing – original draft, Writing – review and editing | Sujatha Srinivasan, Conceptualization, Data curation, Methodology, Supervision, Writing – review and editing | Samuel O. Purvine, Data curation, Methodology, Software, Supervision, Writing – review and editing | Tina L. Fiedler, Data curation, Investigation, Methodology, Writing – review and editing | Owen P. Leiser, Conceptualization, Writing – review and editing | Sean C. Proll, Data curation, Methodology, Software, Supervision, Visualization, Writing – review and editing | Samuel S. Minot, Methodology, Software, Supervision, Visualization, Writing – review and editing | Brooke L. Deatherage Kaiser, Conceptualization, Data curation, Funding acquisition, Methodology, Project administration, Resources, Supervision, Writing – review and editing | David N. Fredricks, Conceptualization, Funding acquisition, Methodology, Project administration, Resources, Supervision, Writing – review and editing

## DATA AVAILABILITY

Database files, other protein sequence files, and metagenomic sequencing data are available at figshare. Code files and metaproteomic search results are available at Zenodo.

## ADDITIONAL FILES

The following material is available online.

Supplemental Material

**Fig. S1 (mSystems.00678-22-s0001.tif).** Percent of peptides identified only in the first replicate of the samples, only in the second replicate, or in both replicates. In a preliminary analysis, spectra from the first and second replicates of the samples were searched against 16S_Sample-Matched databases. The percent of unique peptide sequences associated with a PSM where $q < 0.01$ identified in only one of the replicates, or both, are shown in the figure.

**Fig. S2 (mSystems.00678-22-s0002.tif).** Beta diversity and sample rarefaction curves by 16S rRNA gene sequencing and shotgun metagenomic sequencing. Beta diversity of the 29 samples visualized on multidimensional scaling plots using Bray-Curtis distance for 16S rRNA gene sequencing data (A) and shotgun metagenomic sequencing data (B). IDs for each sample are overlaid and BV status is shown with a coral dot (BV-negative) or a teal dot (BV-positive). Rarefaction curves for the samples were generated by randomly sampling bacterial DNA reads from 16S rRNA gene sequencing (C) and shotgun metagenomic sequencing (D) and tracking new species. Curves are color coded by BV status with a coral line for BV-negative and teal line for BV-positive.

**Fig. S3 (mSystems.00678-22-s0003.tif).** Number of significant fungal PSMs identified by Global and 16S_Pooled database searches. Number of significant fungal PSMs identified in the subset of six samples by searches of the Global and 16S_Pooled databases. Each point represents the number of spectra for a single sample.

**Fig. S4 (mSystems.00678-22-s0004.tif).** Comparison of database performance across all samples. All 29 CVL samples were searched against the five listed database types and one-sided Wilcoxon signed-rank tests were performed to determine whether the database listed on the row generated significantly more human (A) or bacterial (B) PSMs than the database listed under the column. The *P*-value for each test is shown in the cell. Comparisons that were significant ($P < 0.01$) are shaded orange, while nonsignificant comparisons are shaded blue.

**Fig. S5 (mSystems.00678-22-s0005.tif).** Overlap of spectra identified in searches of 16S_Sample-Matched and Shotgun_Sample-Matched databases. Percent of bacterial spectra in each sample identified only by searching its 16S_Sample-Matched database (green), Shotgun_Sample-Matched database (Purple), or identified in both database searches (red).

**Table S1 (mSystems.00678-22-s0006.docx).** Bacterial read counts and alpha diversity from 16S rRNA gene sequencing and shotgun metagenomic sequencing. Number of reads of bacterial DNA and alpha diversity as measured by Shannon index of each sample by both 16S rRNA gene sequencing and shotgun metagenomic sequencing. Shannon index of zero indicates a sample where only one species was detected.

**Table S2 (mSystems.00678-22-s0007.docx).** Number of significant PSMs per sample identified by searches of different protein databases. Number of significant human and bacterial PSMs generated in each sample by database type.

**Table S3 (mSystems.00678-22-s0008.docx).** Percentage of significant PSMs per sample identified to specific taxa by searches of different protein databases. Average percent of bacterial PSMs matched to proteins from only one genus or species by searches of the listed database type.

**Table S4 (mSystems.00678-22-s0009.docx).** Taxa associated with spectra identified in 16S_Sample-Matched database search and missed by Shotgun_Sample-Matched search, and vice-versa. The number of spectra across all samples that were matched to each taxonomic identifier and were identified by the 16S_Sample-Matched search and missed by searching the corresponding Shotgun_Sample-Matched search, or vice versa.

**Table S5 (mSystems.00678-22-s0010.docx).** Comparison of present techniques with past investigations of the vaginal metaproteome in terms of identified human and bacterial proteins, samples analyzed, and database type. Comparison of the number of unique proteins identified in studies of the vaginal metaproteome, past investigations, and the current study. a. Number of samples analyzed. b. The study did not report the number of this type of proteins identified.

## Open Peer Review

**PEER REVIEW HISTORY (review-history.pdf).** An accounting of the reviewer comments and feedback.

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
