## [Reviewer comments · mSystems]

Optimizing Metaproteomics Database Construction: Lessons from a Study of the Vaginal Microbiome

Elliot Lee, Sujatha Srinivasan, Samuel Purvine, Tina Fiedler, Owen Leiser, Sean Proll, Samuel Minot, Brooke Deatherage Kaiser, and David Fredricks

Corresponding Author(s): David Fredricks, Fred Hutchinson Cancer Research Center

Review Timeline:

Submission Date:	July 25, 2022
Editorial Decision:	October 14, 2022
Revision Received:	January 13, 2023
Editorial Decision:	February 13, 2023
Revision Received:	March 21, 2023
Accepted:	April 6, 2023

Editor: Neha Sachdeva

Reviewer(s): Disclosure of reviewer identity is with reference to reviewer comments included in decision letter(s). The following individuals involved in review of your submission have agreed to reveal their identity: Rabindra Kumar Mandal (Reviewer #2)

Transaction Report:

DOI: <https://doi.org/10.1128/msystems.00678-22>

October 14, 2022

Prof. David N Fredricks
Fred Hutchinson Cancer Research Center
Vaccine and Infectious Disease Division
1100 Fairview Avenue North
E4-100
Seattle, WA 98109-1024

Re: mSystems00678-22 (Optimizing Metaproteomics Database Construction: Lessons from a Study of the Vaginal Microbiome)

Dear Prof. David N Fredricks:

Thank you for submitting your manuscript to mSystems. We have completed our review and I am pleased to inform you that, in principle, we expect to accept it for publication in mSystems. However, acceptance will not be final until you have adequately addressed the reviewer comments.

Please review the reviewer comments and revise your manuscript before resubmitting with mSystems Journal.

Preparing Revision Guidelines

Sincerely,

Neha Sachdeva

Editor, mSystems

Journals Department
American Society for Microbiology

Reviewer comments:

Reviewer #1 (Comments for the Author):

Summary:

The authors present a 29-sample case study to share "lessons learned" from designing untargeted metaproteomic reference databases for the vaginal microbiome.

By designing six different databases, using combinations of public reference datasets and sample-specific datasets from translated metagenomic sequencing, the authors attempt to optimize the inherent tradeoff between database size/false discover rates and true hits. The authors find that large datasets lead to high false positive rates, and that hybrid databases allow for more thorough protein identification than reference or sample-specific databases alone.

The authors' aim is to have their findings be applied to any system. They suggest that their findings may be most relevant for other microbial communities with low bacterial density or relatively fewer bacteria to human cells.

Overall comments:

Using a bacterial genome-based databases for untargeted proteomics is bound to lead to numerous false positive hits. Ground-truthing using reference proteomics of bacteria and consortia of interest is highly recommended. This will add specificity to the current approach of adding different number of bacterial genomes from different species.

To substantiate the claims about the generality of these approaches, I would suggest using multiple proteomics datasets from distinct environments and doing a comparative study. For example, the authors may benefit from researching proteomics studies of marine microbial communities or the oral microbiome in addition to the gut.

The authors do not take advantage of the novelty of their samples and potential interesting conclusions relevant to the vaginal microbiome. I would expand upon these findings as well as analogous findings for a few other example datasets, or even simulated metaproteomic data.

In general, the writing is very clear.

In general, figures need to be consolidated and reduced to clarify the major takeaways of the paper. The figures with multiple parts should be formatted as panels. Biorender should be cited in the figure captions of any figure utilizing the service.

More recent references should be used for methods when possible.

Line by line edits:

Line 66: "allows" should be "allow"

Line 106: If race/ethnicity was determined by a survey, this is better written as, "41% (12/29) of study participants identified as Black on the intake survey..."

Line 121: Add reference for 16 contaminants.

Line 130: State what common fungi were included in line

Line 139: Performance evaluation should be heavily weighted on proteins identified based on ground truthing with known references

Lines 186-189: Presenting the differential protein presents and abundances for the BV versus non-BV samples should be highlighted as a result and exemplar of the successful search.

Line 201: A higher false discovery rate is indeed expected for a larger database due to multiple hypothesis testing. How was the q value adjusted as a function of database search space?

Line 281: Add more details of the results comparing the data and databases used here to previous proteomics work in the vaginal microbiome. This result is worthy of its own figure.

Line 295: How was the phylogeny of the reference versus distantly related genomes taken into account for the database depth? Sampling closely related genomes of a species versus distantly related genomes will make a huge difference. As will the size of the flexible genomes (as stated later).

Line 324: How do you expect human PSM generation to vary in different microbiomes where bacteria outnumber human cells by a higher degree?

Line 340: These results should be moved to the results section.

Line 370: Comparisons to a more similar microbiome, like the oral microbiome, should also be made.

Line 381: The relatedness of each genome for each species of interest needs to be included. What about the numerous metagenomically assembled genomes MAGs available for unculturable strains of interest? (see Line 556 also)

Line 429: Can you quantify this observation?

Line 586: Were additional genomes of only *L. crispatus* used or were genomes of each species of interest used?

Table 3: can you quantify the results or show as a schematic?

Fig 3: Use a sample label that has relevance to the reader

Fig. S1: Not needed. Figure is sufficiently summarized in a sentence

Reviewer #2 (Comments for the Author):

Lee et al., have systematically investigated the impact of different databases on identifying peptides, PSMs, from cervicovaginal lavage samples from bacterial vaginosis patients. They have provided recommendations on how to approach metaproteomics studies. Authors need to address the following comments before publication.

Comments:

1. Why were different samples used for DNA sequencing and metaproteomics? Cervicovaginal lavage sample could have been used for DNA sequencing, which might have been better for the entire study.
2. What is the genetic diversity of the vaginal microbiome? Typically, the mouse gut microbiome roughly contains around >5.8 million non-redundant genes (Zhu et al., 2021). Please show gene accumulation curve from shotgun metagenomics.
3. What is the basis for excluding bacteria with relative abundance <0.1%? Is this an arbitrary number?
4. What are the biological significance, usability, and interpretability of identifying just below 1500 unique proteins out of thousands of proteins?
5. Results are not summarized from 16s rRNA gene sequencing and metagenome sequencing. I would show at least alpha and beta diversity for taxonomic assignments from both approaches. Please also mention the sequencing depth of both approaches. Rarefaction plot should justify.
6. What percent of proteomics data can be tied to the specific taxa?
7. Line 84: Increasing database size can also increase the likelihood of more hits.
8. Line 112: Is stochasticity of metaproteomic analysis common phenomenon? Fig. S1 should be included in the main figures.
9. Line 142-46: Why are 16S databases larger than shotgun databases? Shouldn't it be the opposite? Please discuss.
10. Line 218019: What happens to a PSM if it matched to two or more proteins in the database with the same score? Does it discard those PSMs or keep one or all proteins? Is it possible that in a larger dataset, it matches more than one protein, so tosses it?
11. Line 236: Adding additional strains to a database does not seem to improve performance. Figure 8B. shows a similar decreasing trend for both additional Strain and Species. But Species have a higher magnitude of change.

Optimizing Metaproteomics Database Construction: Lessons from a Study of the Vaginal Microbiome

Lee et al., have systematically investigated the impact of different databases on identifying peptides, PSMs, from cervicovaginal lavage samples from bacterial vaginosis patients. They have provided recommendations on how to approach metaproteomics studies. Authors need to address the following comments before publication.

Comments:

1. Why were different samples used for DNA sequencing and metaproteomics? Cervicovaginal lavage sample could have been used for DNA sequencing, which might have been better for the entire study.
2. What is the genetic diversity of the vaginal microbiome? Typically, the mouse gut microbiome roughly contains around >5.8 million non-redundant genes (Zhu et al., 2021). Please show gene accumulation curve from shotgun metagenomics.
3. What is the basis for excluding bacteria with relative abundance <0.1%? Is this an arbitrary number?
4. What are the biological significance, usability, and interpretability of identifying just below 1500 unique proteins out of thousands of proteins?
5. Results are not summarized from 16s rRNA gene sequencing and metagenome sequencing. I would show at least alpha and beta diversity for taxonomic assignments from both approaches. Please also mention the sequencing depth of both approaches. Rarefaction plot should justify.
6. What percent of proteomics data can be tied to the specific taxa?
7. Line 84: Increasing database size can also increase the likelihood of more hits.
8. Line 112: Is stochasticity of metaproteomic analysis common phenomenon? Fig. S1 should be included in the main figures.
9. Line 142-46: Why are 16S databases larger than shotgun databases? Shouldn't it be the opposite? Please discuss.
10. Line 218019: What happens to a PSM if it matched to two or more proteins in the database with the same score? Does it discard those PSMs or keep one or all proteins? Is it possible that in a larger dataset, it matches more than one protein, so tosses it?
11. Line 236: Adding additional strains to a database does not seem to improve performance. Figure 8B. shows a similar decreasing trend for both additional Strain and Species. But Species have a higher magnitude of change.

Reviewer comments:

Author responses to reviewer comments are written in bold.

We would like to thank the reviewers for their thoughtful comments which have helped improve the quality of this manuscript. See point-by-point responses to specific feedback below.

Reviewer #1 (Comments for the Author):

Summary:

The authors present a 29-sample case study to share "lessons learned" from designing untargeted metaproteomic reference databases for the vaginal microbiome.

By designing six different databases, using combinations of public reference datasets and sample-specific datasets from translated metagenomic sequencing, the authors attempt to optimize the inherent tradeoff between database size/false discover rates and true hits. The authors find that large datasets lead to high false positive rates, and that hybrid databases allow for more thorough protein identification than reference or sample-specific databases alone.

The authors' aim is to have their findings be applied to any system. They suggest that their findings may be most relevant for other microbial communities with low bacterial density or relatively fewer bacteria to human cells.

Overall comments:

Using a bacterial genome-based databases for untargeted proteomics is bound to lead to numerous false positive hits. Ground-truthing using reference proteomics of bacteria and consortia of interest is highly recommended. This will add specificity to the current approach of adding different number of bacterial genomes from different species.

To substantiate the claims about the generality of these approaches, I would suggest using multiple proteomics datasets from distinct environments and doing a comparative study. For example, the authors may benefit from researching proteomics studies of marine microbial communities or the oral microbiome in addition to the gut.

The authors do not take advantage of the novelty of their samples and potential interesting conclusions relevant to the vaginal microbiome. I would expand upon these findings as well as analogous findings for a few other example datasets, or even simulated metaproteomic data.

In general, the writing is very clear.

In general, figures need to be consolidated and reduced to clarify the major takeaways of the paper. The figures with multiple parts should be formatted as panels. Biorender should be cited in the figure captions of any figure utilizing the service.

This is an excellent suggestion. The multi-part figures 1, 2, 3, 4, 8, S2, and S4 have been consolidated into panels. We have also cited Biorender in the caption of Figure 1 where we used the service.

More recent references should be used for methods when possible.

The oldest papers referenced in the Methods section are Amsel, *et al.* (1983) and Nugent, *et al.* (1991). These are the original publications for the Amsel Criteria and Nugent Score still commonly used to diagnose bacterial vaginosis. Although we could use more recent references, we felt it was important to give credit to the scientists who originally created these diagnostic methods.

Line by line edits:

Line 66: "allows" should be "allow"

This change has been incorporated into the updated manuscript.

Line 106: If race/ethnicity was determined by a survey, this is better written as, "41% (12/29) of study participants identified as Black on the intake survey..."

This is a good point, and we have updated the text to "41% (12/29) of study participants were identified themselves as Black and 48% were identified as White (14/29)." (lines 107-108 in tracked changes file)

Line 121: Add reference for 16 contaminants.

We have added a reference to Frankenfield, *et al.* (2022) to address the contaminant proteins (line 125 in tracked changes file). The optimal contaminants to select in a metaproteomics experiment is an open question in the field, and we have updated the supplemental methods to reflect this (lines 761-765 in tracked changes file).

Line 130: State what common fungi were included in line

Thank you for pointing out this omission. We have now listed the fungi included in the database: *Alternaria alternata*, *Candida albicans*, *Nakaseomyces glabrata*, *Candida tropicalis*, *Pichia kudravzevii*, and *Saccharomyces cerevisiae* (lines 136-137 in tracked changes file).

Line 139: Performance evaluation should be heavily weighted on proteins identified based on ground truthing with known references

Ground-truthing would certainly strengthen the results of future metaproteomics studies by providing more accurate estimates of false discovery rates. However, there are significant logistical hurdles to re-analyzing our samples with spiked-in proteins to provide this ground-truthing. Additionally, we believe that comparisons of our analysis results to the results of past metaproteomic studies of vaginal samples are valid, as none of these studies used ground-truthing when performing protein identification, either.

Lines 186-189: Presenting the differential protein presents and abundances for the BV versus non-BV samples should be highlighted as a result and exemplar of the successful search.

Although it's straightforward to determine whether a given human protein is differentially abundant between groups, due to the heterogeneity of bacterial proteins with identical/similar functions, it is much harder to say that specific bacterial proteins are differentially abundant. Proteins from individual taxa can be compared fairly easily, but since different taxa dominate the vaginal microbiome in BV- versus BV+ samples, comparing bacterial proteins in this way would primarily recapitulate taxonomic differences. Therefore, to retain some biological relevance for differences in bacterial proteins, we chose to compare bacterial proteins at the level of functional annotations, and do the same for human proteins to maintain consistency.

Line 201: A higher false discovery rate is indeed expected for a larger database due to multiple hypothesis testing. How was the q value adjusted as a function of database search space?

The q-value for a given PSM is a function of both spectral E-value (quality of match between a sample spectrum and database peptide), and effective database size (number of unique peptides generated by running all database proteins through an *in silico* tryptic digest). Therefore, as database size increases, q-values will become less significant in a linear fashion.

Line 281: Add more details of the results comparing the data and databases used here to previous proteomics work in the vaginal microbiome. This result is worthy of its own figure.

This is an excellent point – comparisons of the number of unique proteins identified by our databases compared with past metaproteomic studies of vaginal samples should have been in the results. The section, “Results from optimized databases are in line with past metaproteomic studies of the vaginal microbiome” has been renamed to “Results from optimized database searches identify a large number of unique proteins while remaining in line with past metaproteomic studies of the vaginal microbiome,” and we have added the following text (lines 289-296 in tracked changes file) to the beginning of this section pointing out the number of unique proteins identified by our databases: “Past metaproteomic studies of vaginal samples have primarily used the Uniprot/TrEMBL database as a reference for their bacterial proteins (Table S7). However, we found that a more tailored approach outperformed large, very broad databases. Our sample-matched databases identified a large number of unique proteins compared to other metaproteomic studies of vaginal samples, relative to the number of samples analyzed (Table S7). Shotgun_Sample-Matched database searches identified the most unique human proteins at 1,182, while Hybrid_Sample-Matched identified the most bacterial proteins at 1,418.” We have also added an extra column to Table S7 detailing the types of databases used by past metaproteomic studies in lieu of another figure.

Line 295: How was the phylogeny of the reference genomes taken into account for the database depth? Sampling closely related genomes of a species versus distantly related genomes will make a huge difference. As will the size of the flexible genomes (as stated later).

Phylogeny is not taken into account for calculating database depth, only the number of genomes available for the species on the NCBI website. The metric therefore provides an approximate estimate as to whether additional metagenomic sequencing will provide a benefit. We have added the following text to the Discussion (lines 413-415 in tracked changes file) to acknowledge this fact: “Although this metric is a rough indicator of the completeness of publicly available sequence data as it does not take the full proteomic diversity of community members into account.”

Line 324: How do you expect human PSM generation to vary in different microbiomes where bacteria outnumber human cells by a higher degree?

Because bacterial proteins are almost always more heterogenous than human proteins, they will generally make up the bulk of a metaproteomic database. Since q-values scale downward with increasing database size, this will push some human PSMs below the threshold for significance. So even in niches where bacterial biomass outweighs human biomass, adding more bacterial proteins to a database will likely reduce human PSMs. We had tangentially addressed this point in the Discussion section, but have added the following text (lines 351-353 in tracked changes file) to more explicitly discuss it: “This will be a challenge for analyzing samples from many body sites, as the heterogeneity present in the bacterial proteome will often be greater than that of the human proteome.”

Line 340: These results should be moved to the results section.

Agreed, we have moved this information to the Results (lines 289-297 in tracked changes file).

Line 370: Comparisons to a more similar microbiome, like the oral microbiome, should also be made.

We concur that applying this method to a different set of samples, for example from a different body site or environmental sample, would strengthen this study. Unfortunately, after an in-depth search, we were unable to find a non-gut microbiome study that had performed metaproteomic analysis along with 16S rRNA and metagenomic sequencing, had made their data available, and had included sufficient metadata with the published data for us to perform a similar analysis. We did not use data from the gut microbiome for comparison, as samples from the gut microbiome are so diverse that our approach would create unfeasibly large databases to search. Hopefully, as the technology is used more widely and more multi-omics studies from low- to mid- microbial biomass studies are published with appropriate associated data, this type of re-analysis will become feasible.

Line 381: The relatedness of each genome for each species of interest needs to be included. What about the numerous metagenomically assembled genomes MAGs available for unculturable strains of interest? (see Line 556 also)

Although incorporating information about genome relatedness and pangenome size into the Database Depth metric would certainly make it more accurate, we only intended the metric to help other researchers estimate how many genomes are available for the species in their sample, and therefore gauge how adequate public-sequence databases would be for their analysis. We have added text to the manuscript to clarify that Database Depth is a rough estimator, rather than a strong predictor of proteome coverage (lines 318-321 and 413-415 in tracked changes file).

Line 429: Can you quantify this observation?

Yes, we have added a note that on average, 2.7% of significant bacterial PSMs matched proteins from taxa at <1% abundance in the sample, but still present in the databases (lines 459-460 in tracked changes file).

Line 586: Were additional genomes of only *L. crispatus* used or were genomes of each species of interest used?

For the Additional Strains databases, only genomes from *L. crispatus* were used, since this is a species of vaginal bacteria with a relatively large pangenome and a relatively large number of genomes available on NCBI. We have added the following text to the relevant methods section (lines 618-622 in tracked changes file) to clarify this: "For the Additional Strains databases, these genomes were chosen from 103 additional strains of *L. crispatus* genomes available in RefSeq release 97. For the Additional Species databases, these genomes were chosen from or 103 random bacterial species of different genera assembled to the scaffold level as part of the Human Microbiome Project."

Table 3: can you quantify the results or show as a schematic?

We agree that this was an inefficient way to show comparisons between our results and past studies. We have updated the table to condense and clarify it (Table 3).

Fig 3: Use a sample label that has relevance to the reader

We have reformatted this figure so that BV status of the samples are displayed on the x-axis along with sample ID numbers, and re-ordered the figures to group samples by BV status.

Fig. S1: Not needed. Figure is sufficiently summarized in a sentence

Since the reviewers disagreed on whether this figure should be removed or moved to the main body of the manuscript, we have decided to keep it in the supplements. We defer to the Editor if they prefer to move the figure.

Reviewer #2 (Comments for the Author):

Lee et al., have systematically investigated the impact of different databases on identifying peptides, PSMs, from cervicovaginal lavage samples from bacterial vaginosis patients. They have provided recommendations on how to approach metaproteomics studies. Authors need to address the following comments before publication.

Comments:

1. Why were different samples used for DNA sequencing and metaproteomics? Cervicovaginal lavage sample could have been used for DNA sequencing, which might have been better for the entire study. **The same set of 29 samples was used for metaproteomics and DNA sequencing. We have added the following text (lines 104-106 in tracked changes file) to clarify this detail.**

2. What is the genetic diversity of the vaginal microbiome? Typically, the mouse gut microbiome roughly contains around >5.8 million non-redundant genes (Zhu et al., 2021). Please show gene accumulation curve from shotgun metagenomics.

Our shotgun sequencing approach identified 443,291 non-redundant bacterial proteins across our 29 samples. A recent effort to create a comprehensive gene catalog of the vaginal microbiome identified 950,000 non-redundant bacterial genes in this body site (“A comprehensive non-redundant gene catalog reveals extensive within-community intraspecies diversity in the human vagina” by Ma, *et al.* 2020). We have also added rarefaction curves for both 16S rRNA sequencing and shotgun metagenomic sequencing, along with read depth for both approaches (Table S1, Figure S2).

3. What is the basis for excluding bacteria with relative abundance <0.1%? Is this an arbitrary number? **We chose this cutoff because we regularly observed PSMs which matched taxa with abundance between 1% and 0.1%, but were concerned there would be so few hits on proteins from bacteria with relative abundance <0.1%, it would not be worthwhile including their protein sequences in our databases. We have added the following text (lines 458-461 in tracked changes file) to address this: “We chose this cutoff because on average 2.7% of significant bacterial PSMs non-exclusively matched at least one taxa with relative abundance between 1% and 0.1% (data not shown).”**

4. What are the biological significance, usability, and interpretability of identifying just below 1500 unique proteins out of thousands of proteins?

It is a fair observation that one of the weaknesses of metaproteomic analysis is the relative shallowness of the data generated – only a fraction of the thousands of unique proteins present in a sample can be identified. Still, these identified proteins can provide insights into biologically relevant functional differences between communities, for example the 2,382 GO functional annotations of proteins that were differentially abundant based on BV status in our samples (Table 2), which we intend to address in a follow-up manuscript. We also tried to contextualize this information by comparing the number of unique proteins identified using our optimized databases to past studies of the vaginal microbiome, which generally required many more samples to identify similar numbers of unique proteins (Table S7).

5. Results are not summarized from 16s rRNA gene sequencing and metagenome sequencing. I would show at least alpha and beta diversity for taxonomic assignments from both approaches. Please also mention the sequencing depth of both approaches. Rarefaction plot should justify.

We agreed this was very reasonable information to include in the manuscript and have added these statistics and rarefaction plots to the supplemental materials (Table S1, Figure S2).

6. What percent of proteomics data can be tied to the specific taxa?

This is an excellent additional point of analysis and we appreciate the suggestion. We have included the average percent of bacterial PSMs each database tied to a single genus and species of bacteria in the updated manuscript (Table S3) and referenced it in the Results section (lines 188-193 in tracked changes file).

7. Line 84: Increasing database size can also increase the likelihood of more hits.

We agree, and we have updated this line to reflect his fact (lines 82-85 in tracked changes file).

8. Line 112: Is stochasticity of metaproteomic analysis common phenomenon? Fig. S1 should be included in the main figures.

Stochasticity in results, especially at the peptide level, is very common in metaproteomic analysis. We have added a reference to Bittremieux, *et al.* (2017) which provides a good review of sources of variability in mass spectrometry (Reference #27, line 115 in tracked changes file). Because there was disagreement between the reviewers on whether Fig. S1 should be removed or promoted to the main body of the manuscript, we have decided to keep it in the supplemental material.

9. Line 142-46: Why are 16S databases larger than shotgun databases? Shouldn't it be the opposite? Please discuss.

Due to the number of genomes available for organisms present in our samples, we were actually not surprised that the 16S databases were larger than the shotgun databases. The Pooled databases represent all the sequences we had available for each protein type – public (1,345,203) and translated metagenomic (443,291). Although metagenomic sequencing provides substantial depth (in our study, identifying more than 400,000 unique bacterial genes in the 29 samples we analyzed), it provides less depth than the protein heterogeneity present in the NCBI database, generated by performing whole-genome sequencing on many isolates of different species. Therefore, the 16S rRNA gene databases will be substantially larger.

10. Line 218019: What happens to a PSM if it matched to two or more proteins in the database with the same score? Does it discard those PSMs or keep one or all proteins? Is it possible that in a larger dataset, it matches more than one protein, so tosses it?

Many spectra matched to more than one protein, often because the peptide sequence our search algorithm matched was present in multiple proteins. MS-GF+ reports all proteins containing the matched peptide sequence. What we did with those multi-match PSMs then differed slightly based on the analysis we were performing. i.e. for analyzing what taxa corresponded to a given spectrum, we looked at all protein matches and their associated taxa (lines 687-689 in tracked changes file). For EggNog functional analysis, we took the first protein match and submitted that sequence for functional annotation. But PSMs that matched more than one protein were never excluded from analysis. We have added the following text to the methods (lines 681-682 in tracked changes file) to reflect this detail: “For PSMs that matched multiple proteins, only the first protein match was queried.”

11. Line 236: Adding additional strains to a database does not seem to improve performance. Figure 8B. shows a similar decreasing trend for both additional Strain and Species. But Species have a higher magnitude of change.

This is an excellent point. We have changed this section header to “Adding proteins to a database from additional strains of a species has a less negative effect than adding proteins from additional species due to greater protein heterogeneity,” to better reflect the results (lines 249-251 in tracked changes file).

February 13, 2023

Prof. David N Fredricks
Fred Hutchinson Cancer Research Center
Vaccine and Infectious Disease Division
1100 Fairview Avenue North
E4-100
Seattle, WA 98109-1024

Re: mSystems00678-22R1 (Optimizing Metaproteomics Database Construction: Lessons from a Study of the Vaginal Microbiome)

Dear Prof. David N Fredricks:

Thank you for submitting your manuscript to mSystems. We have completed our review and I am pleased to inform you that, in principle, we expect to accept it for publication in mSystems. However, acceptance will not be final until you have adequately addressed the following reviewer comments.

1. Please explain how the Relative identification rate was calculated. Since this is not a standard metric, this used ideally be done in the line where it is first mentioned.
2. You have not accounted for the diversity among the publicly available genomes for a given species. The Database Depth metric should include an estimate of pangenome coverage, not just the number of available genomes for a given species.

Preparing Revision Guidelines

Sincerely,

Neha Sachdeva

Editor, mSystems

Journals Department
Reviewer comments:

Reviewer #1 (Comments for the Author):

The authors have addressed most comments by editing the language and adding in caveats where their analysis is limited. A few comments remain:

Relative identification rate calculation should be explained in line when first mentioned as it is not a standard metric.

Not accounting for diversity among the publicly available genomes for a given species is problematic. The Database Depth metric should include an estimate of pangenome coverage, not just the number of available genomes for a given species.

Reviewer comments:

Author responses to reviewer comments are written in bold.

Reviewer #1 (Comments for the Author):

1) Relative identification rate calculation should be explained in line when first mentioned as it is not a standard metric.

We agree. We added text to explain this calculation in-line when it is mentioned (lines 301-304 in the marked-up manuscript).

2) Not accounting for diversity among the publicly available genomes for a given species is problematic. The Database Depth metric should include an estimate of pangenome coverage, not just the number of available genomes for a given species.

We agree that the distance between genomes is an important factor when determining how well a species is represented in the public databases. However, the best approach to estimate pangenome coverage is an open question in the field. We have elected to shift the focus of this section. Rather than creating a generalizable metric for the completeness of public sequence databases, we chose to emphasize that adding protein sequences from shotgun metagenomic sequencing has the largest benefit for samples with a high abundance of species with little genomic information available (lines 336-345 in the marked-up manuscript). We used a calculation similar to the Database Depth metric to illustrate this point, but to take the reviewer's comments on inter-genome diversity into account, we used the number of unique tryptic peptide sequences available for a species rather than the number of genome sequences available (lines 318-335 and Fig. 9 in the marked-up manuscript). Counting unique tryptic peptides available in public repositories has the advantage that it reflects the same data which a metaproteomic search program analyzes, and it takes into account the known inter-strain proteomic diversity that is actually present in public sequence databases. We have also updated the Discussion and Methods sections to reflect these changes (lines 424-444, 455-463, and 724-746 in the marked-up manuscript). We appreciate this suggestion.

April 6, 2023

Prof. David N Fredricks
Fred Hutchinson Cancer Research Center
Vaccine and Infectious Disease Division
1100 Fairview Avenue North
E4-100
Seattle, WA 98109-1024

Re: mSystems00678-22R2 (Optimizing Metaproteomics Database Construction: Lessons from a Study of the Vaginal Microbiome)

Dear Prof. David N Fredricks:

Your manuscript has been accepted, and I am forwarding it to the ASM Journals Department for publication. For your reference, ASM Journals' address is given below. Before it can be scheduled for publication, your manuscript will be checked by the mSystems production staff to make sure that all elements meet the technical requirements for publication. They will contact you if anything needs to be revised before copyediting and production can begin. Otherwise, you will be notified when your proofs are ready to be viewed.

If you would like to submit a potential Featured Image, please email a file and a short legend to msystems@asmusa.org. Please note that we can only consider images that (i) the authors created or own and (ii) have not been previously published. By submitting, you agree that the image can be used under the same terms as the published article. File requirements: square dimensions (4" x 4"), 300 dpi resolution, RGB colorspace, TIF file format.

We recognize that the video files can become quite large, and so to avoid quality loss ASM suggests sending the video file via <https://www.wetransfer.com/>. When you have a final version of the video and the still ready to share, please send it to mSystems staff at msystems@asmusa.org.

Sincerely,

Neha Sachdeva
Editor, mSystems

Journals Department
E-mail: mSystems@asmusa.org